# dPob/EMC is essential for biosynthesis of rhodopsin and other multi-pass membrane proteins in *Drosophila* photoreceptors

Takunori Satoh[1], Aya Ohba[1], Ziguang Liu[2], Tsuyoshi Inagaki[1], Akiko K Satoh[1]*

[1]Graduate School of Integrated Arts and Science, Hiroshima University, Higashi-Hiroshima, Japan; [2]Institute of Animal Husbandry, Heilongjiang Academy of Agricultural Sciences, Harbin, China

**Abstract** In eukaryotes, most integral membrane proteins are synthesized, integrated into the membrane, and folded properly in the endoplasmic reticulum (ER). We screened the mutants affecting rhabdomeric expression of rhodopsin 1 (Rh1) in the *Drosophila* photoreceptors and found that dPob/EMC3, EMC1, and EMC8/9, *Drosophila* homologs of subunits of ER membrane protein complex (EMC), are essential for stabilization of immature Rh1 in an earlier step than that at which another Rh1-specific chaperone (NinaA) acts. dPob/EMC3 localizes to the ER and associates with EMC1 and calnexin. Moreover, EMC is required for the stable expression of other multi-pass transmembrane proteins such as minor rhodopsins Rh3 and Rh4, transient receptor potential, and $Na^+K^+$-ATPase, but not for a secreted protein or type I single-pass transmembrane proteins. Furthermore, we found that dPob/EMC3 deficiency induces rhabdomere degeneration in a light-independent manner. These results collectively indicate that EMC is a key factor in the biogenesis of multi-pass transmembrane proteins, including Rh1, and its loss causes retinal degeneration.

*For correspondence: aksatoh@hiroshima-u.ac.jp

Competing interests: The authors declare that no competing interests exist.

## Introduction

In eukaryotes, most integral membrane proteins are synthesized, integrated into the membrane, and folded properly in the endoplasmic reticulum (ER). Molecular chaperones and folding enzymes are required for the folding of the integral membrane proteins in the ER. A comprehensive approach in yeast to identify genes required for protein folding in the ER identified the ER membrane protein complex (EMC), which comprises six subunits (*Jonikas et al., 2009*). Another report studying the comprehensive interaction map of ER-associated degradation (ERAD) machinery revealed that EMC contains four and three additional subunits in mammals and *Drosophila*, respectively (*Christianson et al., 2011*). The deletions of each emc1–6 gene causes the unfolded protein response (UPR), presumably caused by the accumulation of misfolded proteins (*Jonikas et al., 2009*). Meanwhile, a recent study showed that EMC also facilitates lipid transfer from ER to mitochondria (*Lahiri et al., 2014*).

In photoreceptors, the massive biosynthesis of rhodopsin demands chaperones in the ER. In the vertebrate retina, rhodopsin interacts with the ER degradation enhancing α-mannosidase-like 1 (EDEM1) protein and a DnaJ/Hsp40 chaperone (HSJ1B) (*Chapple and Cheetham, 2003*; *Kosmaoglou et al., 2009*). Meanwhile, in *Drosophila* photoreceptors, rhodopsin 1 (Rh1) sequentially interacts with chaperones calnexin99A (Cnx), NinaA, and Xport before exiting from the ER (*Colley et al., 1991*; *Rosenbaum et al., 2006, 2011*). Defects in rhodopsin biosynthesis and trafficking cause retinal degeneration in both *Drosophila* and humans; more than 120 mutations in the rhodopsin gene are associated with human retinitis pigmentosa (*Mendes et al., 2005*; *Xiong and Bellen, 2013*). The overwhelming majority of these mutations lead to misfolded rhodopsin, which aggregates in the secretory pathway (*Hartong et al., 2006*). Thus, it is important to understand the mechanisms

**eLife digest** The membranes that surround cells contain many proteins, and those that span the entire width of the membrane are known as transmembrane proteins. Rhodopsin is one such transmembrane protein that is found in the light-sensitive 'photoreceptor' cells of the eye, where it plays an essential role in vision.

Transmembrane proteins are made inside the cell and are inserted into the membrane surrounding a compartment called the endoplasmic reticulum. Here, they mature and 'fold' into their correct three-dimensional shape with help from chaperone proteins. Once correctly folded, the transmembrane proteins can be transported to the cell membrane. Incorrect folding of proteins can have severe consequences; if rhodopsin is incorrectly folded the photoreceptor cells can die, leading to blindness in humans and other animals.

Experiments carried out in zebrafish have shown that the chaperone protein Pob is required for the survival of photoreceptor cells. Pob is part of a group or 'complex' of chaperone proteins in the endoplasmic reticulum called the EMC complex. This suggests that the EMC complex may be involved in folding rhodopsin, but the details remain unclear.

Here, Satoh et al. studied the role of the EMC complex in the folding of rhodopsin in fruit flies. This involved examining hundreds of flies that carried a variety of genetic mutations and that also had low levels of rhodopsin. The experiments show that dPob—the fly version of Pob—and two other proteins in the EMC complex are required for newly-made rhodopsin to be stabilized. If photoreceptor cells are missing proteins from the complex, the light-sensitive structures in the eye degenerate.

Rhodopsin is known as a 'multi-pass' membrane protein because it crosses the membrane multiple times. Satoh et al. found that the EMC complex is also required for the folding of other multi-pass membrane proteins in photoreceptor cells. The next challenge will be to reveal how the EMC complex is able to specifically target this type of transmembrane protein.

underlying the folding and trafficking of rhodopsin as well as retinal degeneration caused by misfolded rhodopsin.

In zebrafish the partial optokinetic response b $(pob)^{a1}$ mutant exhibits red cone photoreceptor degeneration (*Brockerhoff et al., 1997*; *Taylor et al., 2005*). The localization of overexpressed zebrafish Pob protein in cultured cells in the early secretory pathway including the ER and Golgi body indicates that Pob is involved in red cone rhodopsin transport (*Taylor et al., 2005*). The zebrafish *pob* gene is the homolog of a subunit of EMC, EMC3. Here we report the function of dPob, *Drosophila pob* homolog, on Rh1 maturation, photoreceptor maintenance, and expression of other multi-pass membrane proteins.

## Results

### dPob is essential for maturation and transport of Rh1

Retinal mosaic screening using the FLP/FRT method and two-color fluorescent live imaging was used to identify the genes essential for Rh1 maturation and transport (*Satoh et al., 2013*). For selected lines exhibiting defects in Rh1 accumulation in the live imaging screening, the immunocytochemical distribution of Rh1 was investigated to evaluate the phenotype with respect to transport and morphogenesis (Table 2, *Satoh et al., 2013*). Among them, $CG6750^{e02662}$ (Kyoto stock number: 114504) exhibits severe Arrestin2::GFP and Rh1 reduction in rhabdomeres (*Figure 1A,C*) with normal ommatidial organization. $CG6750^{e02662}$ has an insertion of a piggyBac transposon right downstream of the stop codon of *CG6750* (*Figure 1B*). The phenotype was reverted by the precise excision of the piggyBac transposon or transgenically-expressed *CG6750* (data not shown); this indicates Rh1 reduction is caused by reduced *CG6750* gene function. *CG6750* shares 65% identity and 82% similarity with zebrafish *pob* and 27% identity and 44% similarity with yeast *EMC3*. Because *CG6750* is likely to be the homolog of zebrafish *pob*, we designated *CG6750* as 'dPob' and analyzed its functions in Rh1 transport and retinal morphogenesis.

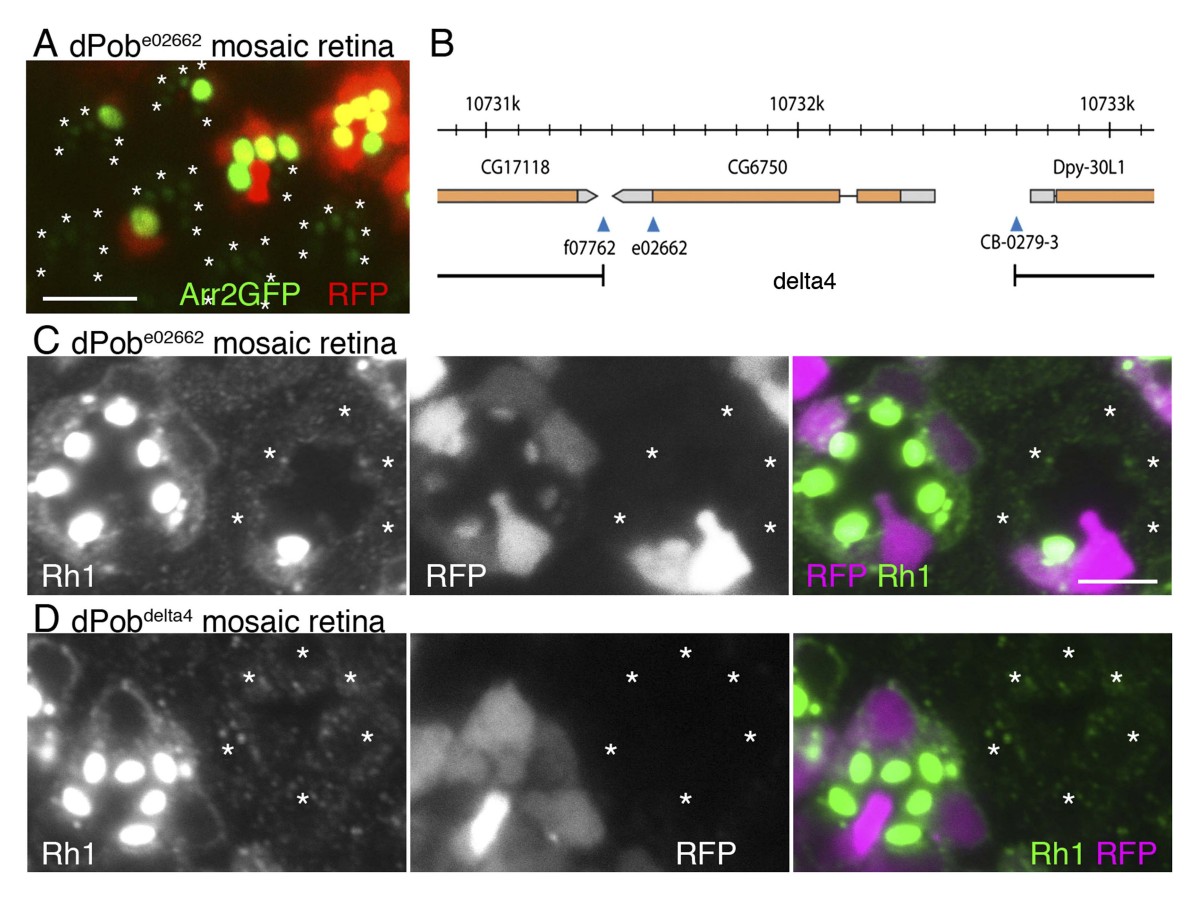

**Figure 1.** Identification of CG6750 as an essential gene for rhodopsin 1 (Rh1) biosynthesis. (**A**) Observation of fluorescent protein localizations in *CG6750^e02662* mosaic retinas by the water immersion technique. RFP (red) indicates wild-type photoreceptors (R1–R8). Arrestin2::GFP (green) shows endogenous Rh1 localization in R1–R6 peripheral photoreceptors. (**B**) Schematic drawing of *CG6750* and insertion/deletion mutants. The *dPob^-* null mutant allele, *dPob^Δ4*, was created by the recombination of two FRTs on *dPob^f07762* and *dPob^CB−0279−3* using an FRT/FLP-based deletion method. (**C, D**) Immunostaining of *dPob^e02662* (**C**) and *dPob^Δ4* (**D**) retinas expressing RFP as a wild-type cell marker (magenta) by anti-Rh1 antibody (green). Asterisks show mutant cells. Scale bar: 5 μm (**A, C, D**).

To address the possibility that the severe reduction of Rh1 protein in *dPob^e02662* mutant is caused by the reduction of mRNA, Rh1 mRNA was quantified in whole-eye clones of the mutant. When compared with control FRT40A whole-eye clone, relative mRNA levels normalized to Act5C were, Rh1: 0.51 (n = 4, S.D. = 0.24); trp: 0.31 (n = 4, S.D. = 0.17); and Arr2: 0.49 (n = 4, S.D. = 0.24). Thus, the great reduction of the Rh1 protein level in *dPob^e02662* clones could not be interpreted by the reduction of mRNA.

As expected from the position of the insertion, dPob was still weakly expressed in *dPob^e02662* homozygous photoreceptors (*Figure 2B,C*), so it was classified as a hypomorphic allele. To further investigate the function of dPob, *dPob^Δ4*, a null mutant allele lacking the entire coding sequence of dPob, was created using an FRT/FLP-based deletion method (*Figure 1B*) (*Parks et al., 2004*). Unlike *dPob^e02662*, which gives escapers up to the late pupal stage, *dPob^Δ4* flies were lethal in the first instar larval stage. Immunostaining of *dPob^Δ4* mosaic retinas shows a great reduction of Rh1 in *dPob^Δ4* homozygous photoreceptors, similar to *dPob^e02662* homozygous photoreceptors (*Figure 1D*).

Next, antisera against dPob (*Figure 2*) were created to investigate dPob localization in fly photoreceptors. Four antisera (three against the N-terminal and one against the C-terminal) recognized a single ~27 kD band in wild-type head homogenates by immunoblotting (*Figure 2A*). This band was greatly reduced in *dPob^e02662* homozygous head homogenates, indicating that these four antisera recognized dPob and that the molecular weight of dPob is ~27 kD. In immunostaining *dPob^e02662* mosaic retinas, two of the C-terminal antisera (dPob-C1 and dPob-C3) produced similar

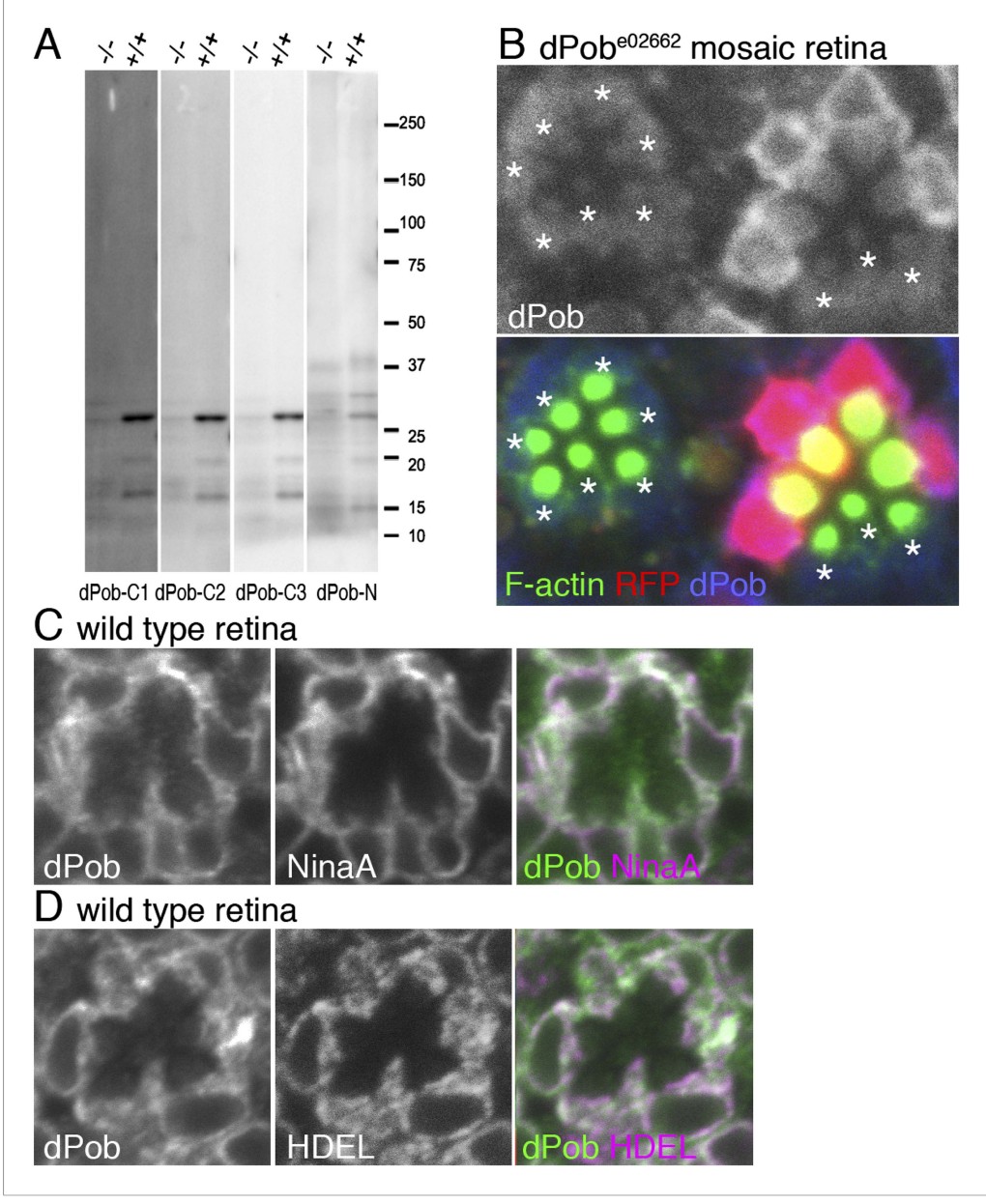

**Figure 2**. Construction of antisera against dPob. (**A**) Immunoblotting of wild-type (+/+) and *dPob^{e02662}* homozygous (−/−) extracts from whole larvae using antiserum against dPob N- and C-terminal polypeptides. (**B**) Immunostaining of a *dPob^{e02662}* mosaic retina expressing RFP (red) as a wild-type cell marker (not shown) by rat anti-dPob-C1 antiserum (blue) and phalloidin (green). Asterisks show *dPob^{e02662}* homozygous photoreceptors. (**C, D**) Immunostaining of wild-type retinas by anti-dPob (green) and anti-NinaA (**C**) or anti-HDEL (**D**) antisera. Scale bar: 5 µm (**B–D**).

staining patterns in the cytoplasm of wild-type cells which were reduced in *dPob^{e02662}* homozygous photoreceptors (*Figure 2B* and *Figure 3B*), indicating that these two antisera recognized dPob in tissue. Because dPob-C3 antiserum had the highest reactivity, we used it in further experiments. Anti-dPob reactivity co-localized with ER markers NinaA and HDEL (*Figure 2C,D*), indicating ER localization of dPob in fly photoreceptors.

## dPob is essential for the biosynthesis of Rh1 apoprotein

Rh1 comprises opsin (an apoprotein) and *11-cis* retinal (a chromophore). Without the chromophore, newly synthesized Rh1 apoprotein accumulates in the ER as an *N*-glycosylated immature form

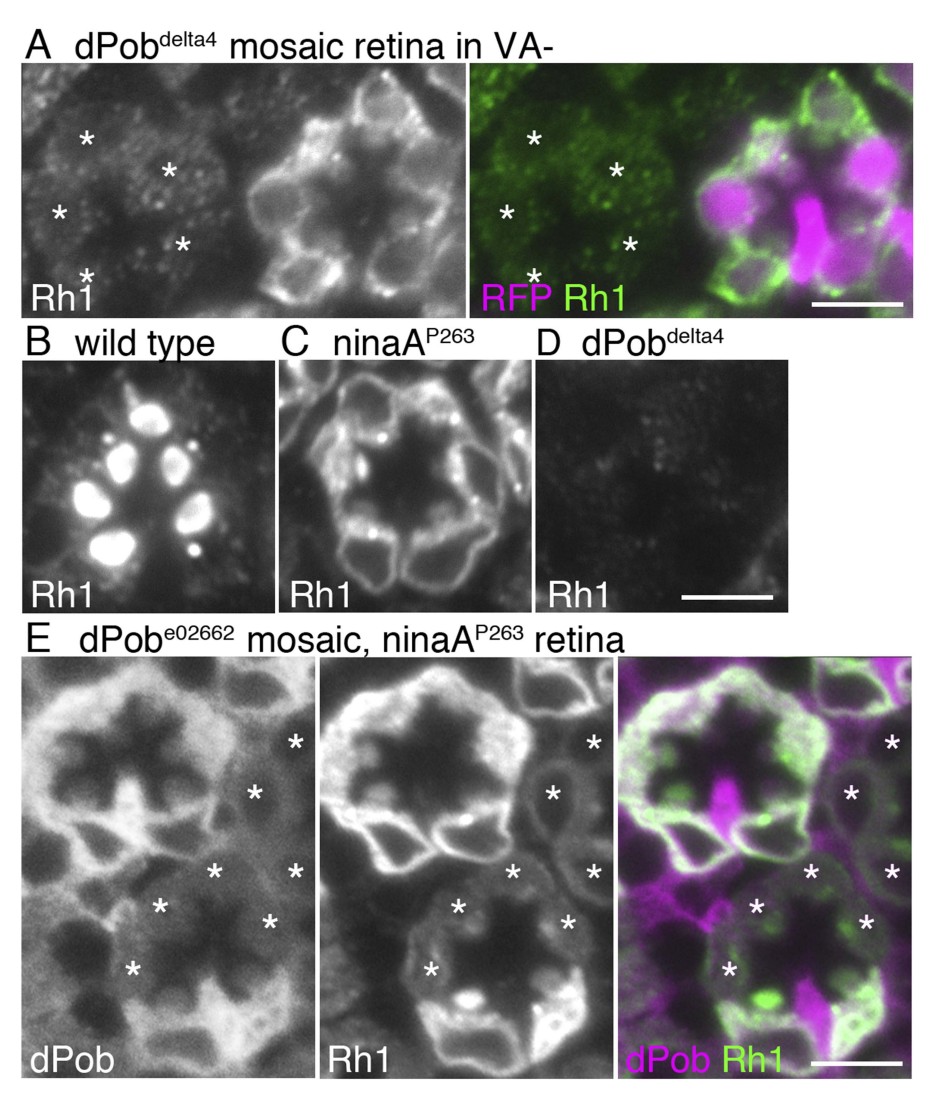

**Figure 3**. dPob stabilizes rhodopsin 1 (Rh1) apoprotein. (**A**) Immunostaining of a *dPob*^Δ4 mosaic retina from a fly reared in vitamin A (VA)-deficient medium by anti-Rh1 antibody. Asterisks show *dPob*^Δ4 homozygous photoreceptors. (**B–D**) Immunostaining of a wild-type (**B**), *ninaA*^P263(**C**), or *dPob*^Δ4 (**D**) ommatidium of flies reared in normal vitamin A-containing medium. (**E**) Immunostaining of a *dPob*^e02662 mosaic retina in *ninaA*^P263 homozygous mutant background from a fly reared in normal medium. Asterisks show *dPob*^Δ4 homozygous photoreceptors. Scale bar: 5 μm (**A–E**).

(*Ozaki et al., 1993*). To investigate whether dPob is essential for the accumulation of Rh1 apoprotein in the ER, *dPob*^Δ4 mosaic retinas were observed in flies reared in medium lacking vitamin A, the source of the chromophore (*Figure 3A*). Rh1 apoprotein was greatly reduced in *dPob*^Δ4 photoreceptor cells, indicating that dPob is essential for the early stage of Rh1 biosynthesis before chromophore binding in the ER.

NinaA, the rhodopsin-specific peptidyl-prolyl-*cis-trans*-isomerase, is a known Rh1 chaperone. In contrast to dPob deficiency, which lacks both Rh1 apoprotein and mature Rh1 (*Figure 3D*), loss of NinaA causes accumulation of Rh1 apoprotein in the ER similar to that observed in the chromophore-depleted condition (*Colley et al., 1991*) (*Figure 3C*). To investigate the epistatic interaction between dPob and NinaA for Rh1 synthesis, Rh1 apoprotein was observed in the *dPob*^Δ4/*ninaA*^P263 double mutant. Rh1 apoprotein was greatly reduced in *dPob*^Δ4/*ninaA*^P263 double-mutant photoreceptors, similar to that in the *dPob*^Δ4 single mutant (*Figure 3E*). This indicates that dPob is epistatic to NinaA.

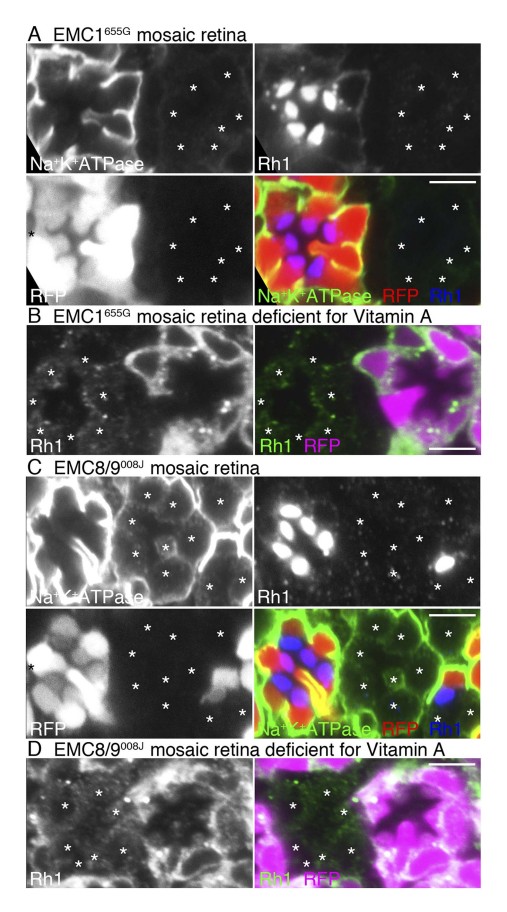

**Figure 4**. Loss of rhodopsin 1 (Rh1) apoprotein in EMC1 and EMC8/9 deficiency. Immunostaining of a *EMC1655G* mosaic retina (**A**, **B**) or a *EMC8/9008J* mosaic retina (**C**, **D**) reared in normal (**A**, **C**) and vitamin A-deficient media (**B**, **D**). Asterisks show *EMC1655G* or *EMC8/9008J* homozygous photoreceptors. RFP (red) indicates wild-type photoreceptors (R1–R8). (**A**, **C**) Na+K+-ATPase, green; Rh1, blue; RFP, red. (**B**, **D**) Rh1, green; RFP, magenta. Scale bar: 5 μm (**A**–**D**).

Cnx is also an Rh1 chaperone and is known to be epistatic to NinaA. Rh1 apoprotein is greatly reduced in both the *cnx1* mutant and *cnx1/ ninaAp269* double mutant (*Rosenbaum et al., 2006*), suggesting that dPob functions in the same stage or a stage close to that in which Cnx functions.

## Other mutants with dPob-like phenotype

The null mutant of dPob shows a characteristic phenotype with no detectable protein expression of Rh1 and very weakened expression of other multiple-transmembrane domain proteins such as Na+K+-ATPase in the mosaic retina (see below). We did not find any other mutant lines with such a phenotype in the course of mosaic screening among 546 insertional mutants described previously (*Satoh et al., 2013*). To explore other mutants showing phenotypes similar to the dPob null mutant, we examined a collection of 233 mutant lines deficient in Rh1 accumulation in photoreceptor rhabdomeres obtained in an ongoing ethyl methanesulfonate (EMS) mutagenesis screening. The detail of the screening will be published elsewhere; at present the Rh1 accumulation mutant collection covers three chromosome arms, approximately 60% of the *Drosophila melanogaster* genome. Under the assumption of a Poisson distribution of the mutants on genes, the collection stochastically covers more than 80% of genes in those arms. The distribution of Rh1 and Na+K+-ATPase was examined for 55 lines of mutants on the right arm of the third chromosome, 93 lines of mutants on the right arm of the second chromosome, and 85 mutants on the left arm of the second chromosome. Among them, only two lines—665G on the right arm of the third chromosome and 008J on the right arm of the second chromosome—showed a dPob null-like phenotype in the mean distribution of Rh1 and Na+K+-ATPase in the mosaic retina (*Figure 4A,C*).

Meiotic recombination mapping and RFLP analysis (*Berger et al., 2001*) were used to map the mutations responsible for the dPob-like phenotype of 008J and 655G. Close linkage of the mutation responsible for the dPob-like phenotype of 655G indicated that the responsible gene is located close to the proximal FRT. Since CG2943 gene, the potential *Drosophila* homolog of EMC1, is also close to the proximal FRT, CG2943 was recognized as a candidate of the responsible gene of 655G. As expected, Df(3R)BSC747, which is lacking the CG2943 gene, failed to complement the lethality of 655G. Targeted re-sequencing in the vicinity of CG2943 revealed that 655G has a two-base deletion at 3R:3729838-3729839 which causes a frame-shift mutation of CG2943, causing185aa deletion from I730 to C-terminus adding polypeptide of RTVRGQESGKQQCLEFLASSANAPRGAPVLYTAHNS. The only membrane-spanning helix of CG2943 is lost in this frame-shift mutation.

RFLP analysis narrowed down the cytology of the responsible gene of 008J to 58D2−59D11. Whole genome re-sequencing revealed that the 008J chromosome obtained three unique mutations in the mapped region compared with the starter stock: one silent mutation on CG30274 at 2R:18714026, a missense mutation on MED23 (E329K) at 2R:18777637, and one nonsense mutation on CG3501 at

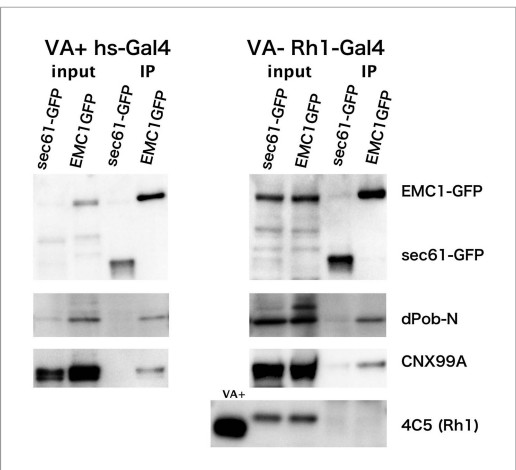

**Figure 5**. Co-immunoprecipitation of EMC1::GFP with dPob and calnexin (Cnx). Immunoblotting of precipitates with anti-GFP antibody from the head extract was prepared from Rh1-Gal4/UAS-EMC1::GFP or sec61:: GFP flies reared in a vitamin A (VA)-deficient medium (left) or heat shock (hs)-Gal4/UAS-EMC1::GFP or sec61:: GFP flies reared in a vitamin A-containing normal medium (right). The mature form of rhodopsin 1 (Rh1) is accumulated in the rhabdomeres in normal medium but not in vitamin A-deficient medium. Instead of the mature form, an N-glycosylated immature form of Rh1 with a larger molecular weight accumulated in the endoplasmic reticulum of flies reared in the vitamin A-deficient medium. In both input extracts prepared from Rh1-Gal4/UAS-EMC1::GFP or sec61::GFP flies there is a band with the same position as EMC1GFP; this band will be the protein cross-reacting to anti-GFP antibody.

2R:18770005 which turns Q40 to a stop codon. Complementation with the deficiencies over the MED23 (BSC783, BSC784) excluded the missense mutation on MED23 from the candidate mutation responsible for the dPob-like phenotype. The amino acid sequence of CG3501 shows 38% and 39% identity to the human EMC8 and EMC9, respectively, and no other gene similar to EMC8/9 was found in the *Drosophila* genome. Based on these results, we identified 655G and 008J as a loss of functional mutation of EMC1 and EMC8/9 of *Drosophila* and named these alleles $EMC1^{655G}$ and $EMC8/9^{008J}$.

We investigated whether EMC1 and EMC8/9 are necessary for the accumulation of Rh1 apoprotein in the ER using $EMC1^{655G}$ and $EMC8/9^{008J}$ mosaic retinas reared in medium lacking vitamin A (*Figure 4B,D*). Rh1 apoprotein was greatly reduced in both $EMC1^{655G}$ and $EMC8/9^{008J}$ photoreceptor cells, indicating that EMC1 and EMC8/9 are also essential for the early stage of Rh1 biosynthesis, like dPob.

## EMC1 binds to dPob and Cnx

To investigate if EMCs form a complex and bind to Rh1 apoprotein, we performed a co-immunoprecipitation assay (*Figure 5*). Since C-terminally tagged dPob protein did not predominantly localize to the ER in vivo (data not shown), GFP-tagged EMC1 protein (EMC:: GFP) was used as the bait. A protein-trap line expressing GFP-tagged sec61alpha protein (sec61::GFP) which localizes in the ER membrane was used as a negative control. Since the overall expression level of EMC1::GFP was strong, hs-Gal4 driver was used to activate UAS:EMC1::GFP for most of the experiments. To analyze the interaction between EMC1 and Rh1 apoprotein, Rh1-Gal4 driver was also used because the expression of EMC1::GFP was stronger in the photoreceptors (data not shown). For the Rh1-Gal4 experiment, flies were reared in a medium lacking vitamin A to accumulate Rh1 apoprotein in the ER. Membrane fraction was recovered from the adult heads, the membrane proteins extracted by CHAPS from the adult head membrane fraction were bound to anti-GFP magnetic beads, and the elutions were analyzed by immunoblotting with antibodies against GFP, Rh1, dPob, and Cnx.

EMC1::GFP and sec61::GFP were concentrated in the immunoprecipitated extract from flies expressing either in the photoreceptor or in the whole head. dPob was co-immunoprecipitated with EMC1::GFP much more strongly than with sec61::GFP. Cnx was also well co-immunoprecipitated with EMC1::GFP but was barely detectable with sec61::GFP. However, Rh1 was not co-immunoprecipitated with EMC1::GFP from vitamin A-deficient photoreceptors accumulating immature Rh1 apoprotein in the ER. These results indicate that dPob and EMC1 are in a complex in vivo, as shown in yeast, and Cnx can also be associated with the complex, which is consistent with the result of epistatic analysis; the stage at which dPob works on the expression of Rh1 apoprotein is close to that of Cnx. Despite the requirement for the expression of Rh1 and co-localization with immature Rh1 apoprotein in the ER, EMC1 does not stably bind to Rh1, indicating that the EMC complex is only temporarily associated with Rh1 apoprotein.

## EMC/dPob is required for the expression of multi-pass membrane proteins

To investigate the substrate specificity of EMC/dPob, we investigated the expressions of secreted or transmembrane proteins other than Rh1 in $dPob^{\Delta 4}$ mosaic retinas. In $dPob^{\Delta 4}$ photoreceptors, multi-pass membrane proteins, the alpha subunit of $Na^{+}K^{+}$-ATPase (*Figure 6A*) and transient receptor potential

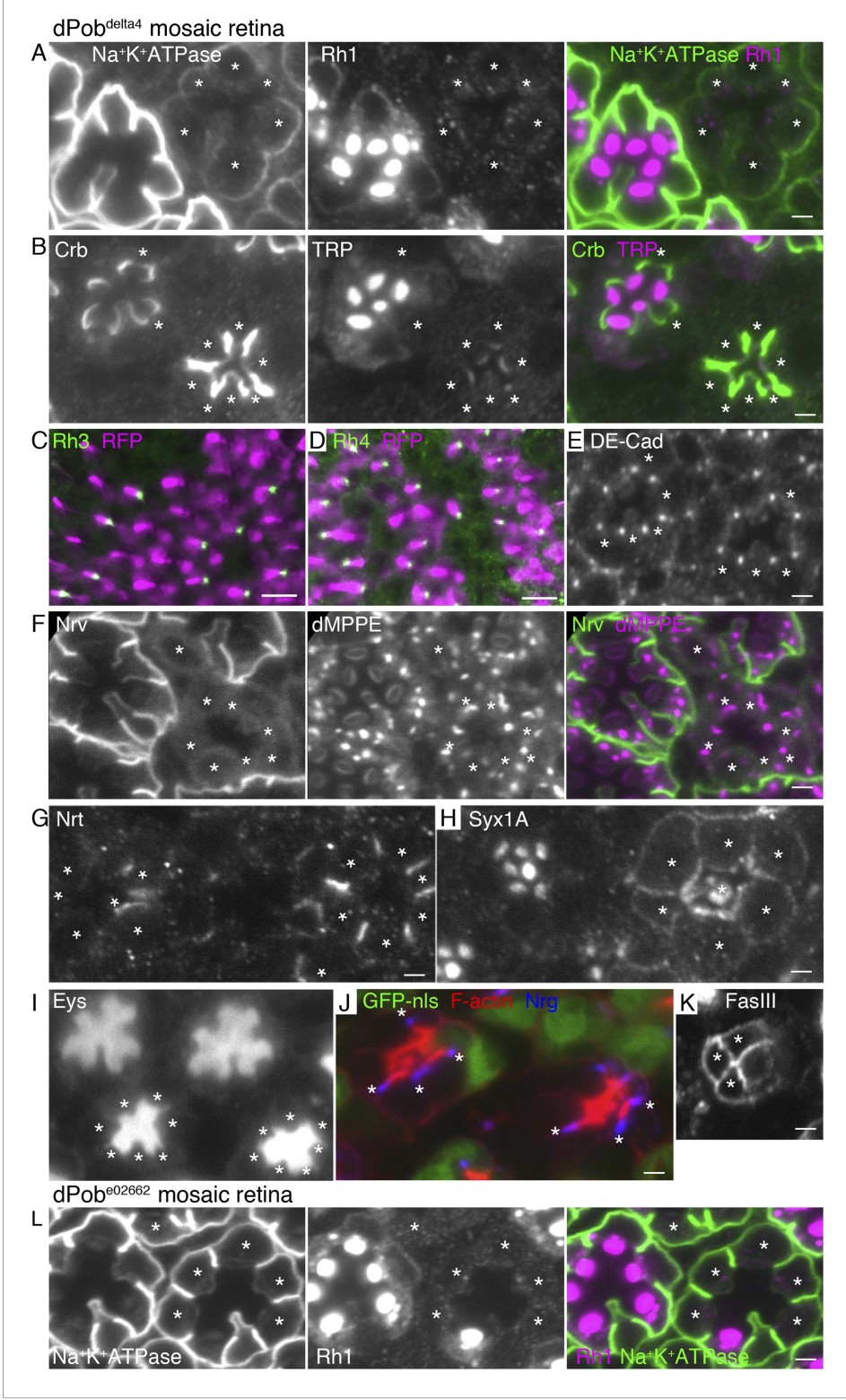

**Figure 6**. Essential role of dPob in the biosynthesis of multi-pass transmembrane proteins. Immunostaining of a *dPob^{Δ4}* mosaic retina (**A–H**) or a *dPob^{e02662}* mosaic retina (**I**). Asterisks show *dPob* homozygous photoreceptors. (**A**) Na+K+-ATPase, green; Rh1, magenta. (**B**) Crb, green; TRP1, magenta. (**C, D**) Rh3 (**C**) and Rh4 (**D**), green; RFP (wild-type cell marker), magenta. Although the boundary between dPob^{Δ4} and wild-type cells is unclear, all green signals are attached to RFP-expressing cell bodies, indicating that mutant R7 cells do not express Rh3 (**C**) or Rh4 (**D**).
*Figure 6. continued on next page*

*Figure 6. Continued*

(**E**) DE-Cad staining. (**F**) Nrv, the beta subunit of Na$^+$K$^+$-ATPase, green; dMPPE, magenta. (**G**) Nrt staining. (**H**) Syx1A staining. (**I**) Eys staining. (**J**) Nrg, blue; F-actin, red; GFP-nls (wild-type cell marker), green. (**K**) FasIII staining. (**L**) Na$^+$K$^+$-ATPase, green; Rh1, magenta. Scale bar: 2 μm (**A**, **B**), 10 μm (**C**, **D**), 2 μm (**E–I**).

(TRP) (*Figure 6B*), were greatly reduced and neither anti-Rh3 nor anti-Rh4 staining was detected (*Figure 6C,D*). On the other hand, the type I single-pass membrane proteins Crb (*Figure 6B*) and DE-Cad (*Figure 6E*) were localized normally in the stalks and adherence junctions in *dPob$^{Δ4}$* photoreceptors. Similarly, a type II single-pass membrane protein Nrt (*Figure 6G*) and a type VI single-pass membrane protein Syx1A (*Figure 6H*) were localized normally in Golgi units and on the plasma membrane in *Pob$^{Δ4}$* photoreceptors. Eys, a secreted protein that expands the inter-rhabdomeric space (IRS) (*Husain et al., 2006*; *Zelhof et al., 2006*), was also secreted normally in *dPob$^{Δ4}$* ommatidia, as expected from the near-normal size of the IRS (*Figure 6I*). Two other type I single-pass membrane proteins expressed in retinal cone cells, Neuroglian (Nrg) and Fasiclin III (FasIII), exhibited normal localization in contact sites between cone cells and cone cell feet (*Figure 6J,K*). Only one type II single-pass membrane protein, the beta subunit of Na$^+$K$^+$-ATPase (Nrv), showed deficient expression in *Pob$^{Δ4}$* photoreceptors (*Figure 6F*). As alpha and beta subunits of Na$^+$K$^+$-ATPase are assembled into a heterodimer within the ER and then transported to the plasma membrane, the absence of Nrv in *Pob$^{Δ4}$* photoreceptors can be interpreted as a consequence of the lack of the multi-pass alpha subunit. These results indicate that dPob is essential for the normal biosynthesis of multi-pass membrane proteins but not for single-pass membrane proteins or secreted proteins.

*EMC1$^{655G}$*- and *EMC8/9$^{008J}$*-deficient photoreceptors show similar substrate specificity to *dPob$^{Δ4}$*-deficient photoreceptors (*Figure 6* and *Figure 7*). In both mutants, accumulation of the membrane proteins with multiple transmembrane domains, Na$^+$K$^+$-ATPase (*Figure 4A,C*), Rh3, Rh4 and TRP (*Figure 7A,C*), on the plasma membrane are greatly reduced in the photoreceptors. However, a type I single-pass transmembrane protein, Crb, is localized intensively in the stalks in *EMC1$^{655G}$* or *EMC8/9$^{008J}$* mutant photoreceptors (*Figure 7B,D*). A type II single-pass membrane protein, Nrt, and a type VI single-pass membrane protein, Syx1A, is localized normally in Golgi units and on the plasma membrane in *EMC1$^{655G}$* and *EMC8/9$^{008J}$* photoreceptors, respectively (*Figure 7C,F*). Eys was also secreted normally and formed a near-normal size of IRS in *EMC1$^{655G}$* or *EMC8/9$^{008J}$* mutant ommatidia (*Figure 7B,D*). Similar to *Pob$^{Δ4}$* photoreceptors, a type II single-pass membrane protein, the beta subunit of Na$^+$K$^+$-ATPase (Nrv) was not detected in the plasma membrane of *EMC1$^{655G}$* or *EMC8/9$^{008J}$* photoreceptors (data not shown).

We observed the expression of dMPPE (*Cao et al., 2011*), a Golgi luminal metallophosphoesterase, anchored by a type II transmembrane helix in the N-terminal region and another transmembrane helix in the C-terminal region. dMPPE was expressed normally in *Pob$^{Δ4}$*, *EMC1$^{655G}$*, and *EMC8/9$^{008J}$* mutant photoreceptors (*Figures 6F, 7C,F*). As two transmembrane helices of dMPPE are separated from each other by the enzymatic domain, these two helices might not associate but behave as two separate transmembrane helices. The EMC requirement for proteins with two transmembrane helices therefore remains unclear.

## ER membrane amplification in dPob-deficient photoreceptors

Electron microscopic observation of thin sections of late pupal flies showed massive amplification of the ER membrane in both *dPob$^{e02662}$* and *dPob$^{Δ4}$* photoreceptors (*Figure 8A–C*) despite the substantial reduction in immature Rh1 apoprotein. In *dPob$^{e02662}$* photoreceptors the ER maintains its sheet structures: the number and length of the sheets was greatly increased but their lumens were almost normal with slight swelling and the sheets were aligned at a regular distance. Meanwhile, in *dPob$^{Δ4}$* photoreceptors the ER sheet structures were no longer maintained and the cytoplasmic space was filled with ER membrane with a larger luminal space. Golgi bodies were also swollen and dilated, and sometimes vesiculated (*Figure 8A–C*, insets). Moreover, concordant with the reduction in Rh1, the rhabdomeres in dPob mutant photoreceptors were quite small and thin but the adherence junctions and basolateral membrane exhibited normal morphology. ER membrane amplification and rhabdomere membrane reduction therefore represent the most prominent phenotype in *dPob*-deficient photoreceptors.

The massive amplification of the ER membrane in both *dPob$^{e02662}$* and *dPob$^{Δ4}$* photoreceptors prompted us to quantify the amounts of residual ER proteins using anti-KDEL and HDEL antibodies.

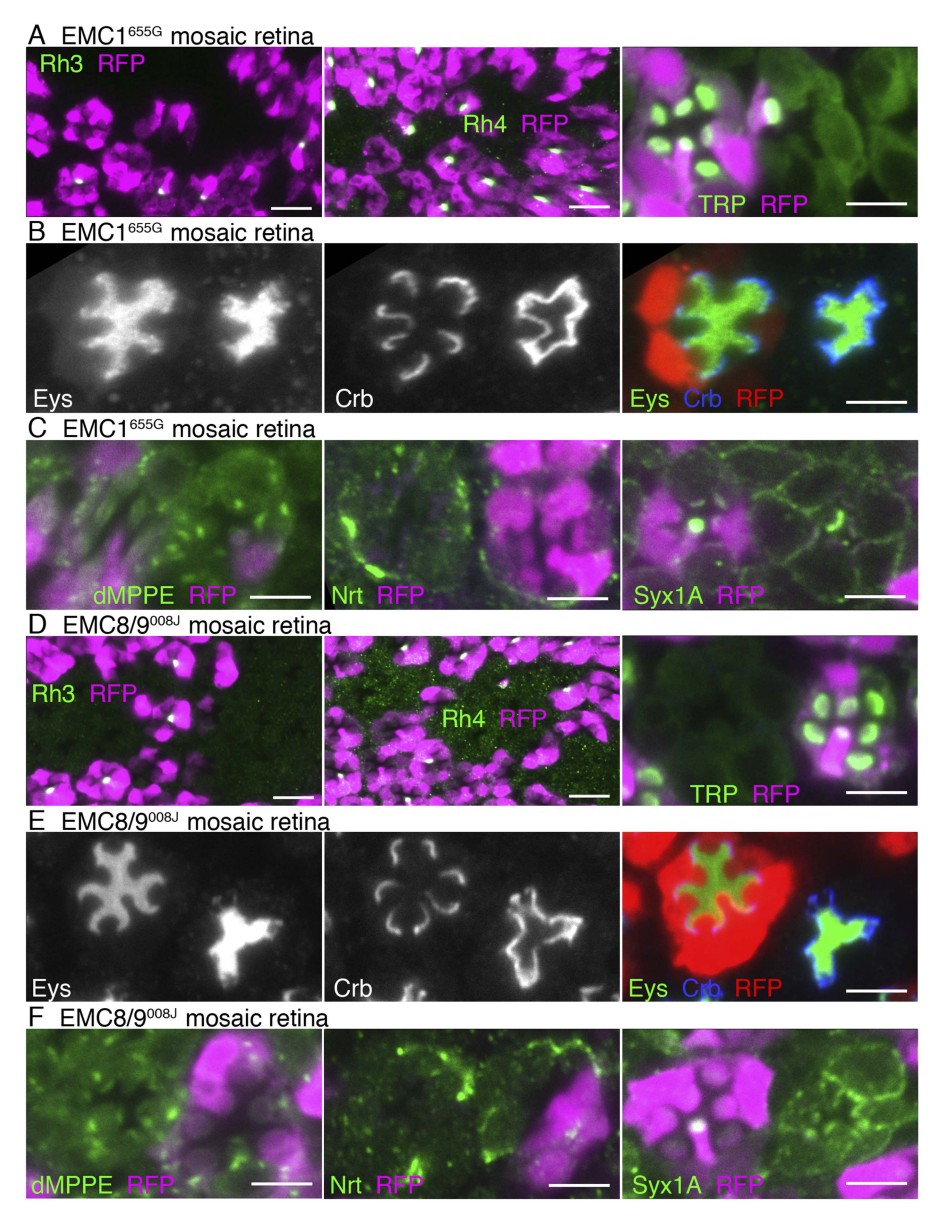

**Figure 7**. Essential role of EMC1 and EMC8/9 in the biosynthesis of multi-pass transmembrane proteins. Immunostaining of a *EMC1^655G* mosaic retina (**A**, **B**, **C**) or a *EMC8/9^008J* mosaic retina (**D**, **E**, **F**). (**A**, **D**) Left: Rh3, middle: Rh4, right: TRP in green, RFP in magenda. (**B**, **E**) Eys in green, Crb in blue, and RFP, wild-type cell marker in red. (**C**, **F**) Left: dMPPE, middle: Nrt, right: Syx1A in green, RFP in magenda. Scale bar: 10 μm (left and middle in **A**, **D**), 5 μm (right in **A**, **D**), 5 μm (**B**, **C**, **E**, **F**).

KDEL and HDEL sequences are signals for ER retention, and *Drosophila* ER resident chaperones including Hsp70–3 and PDI contain these sequences (*Bobinnec et al., 2003*; *Ryoo et al., 2007*). Corresponding to ER membrane amplification, anti-HDEL and anti-KDEL staining were greatly increased in dPob-deficient photoreceptors (*Figure 8D,E*).

## Upregulated unfolded protein responses in dPob-deficient photoreceptors

Accumulation of unfolded proteins in the ER invokes the UPR, which includes activation of the transcription of chaperones and related genes, suppression of translation and enhanced degradation of unfolded protein. The UPR is regulated by some unique intracellular signal transduction pathways.

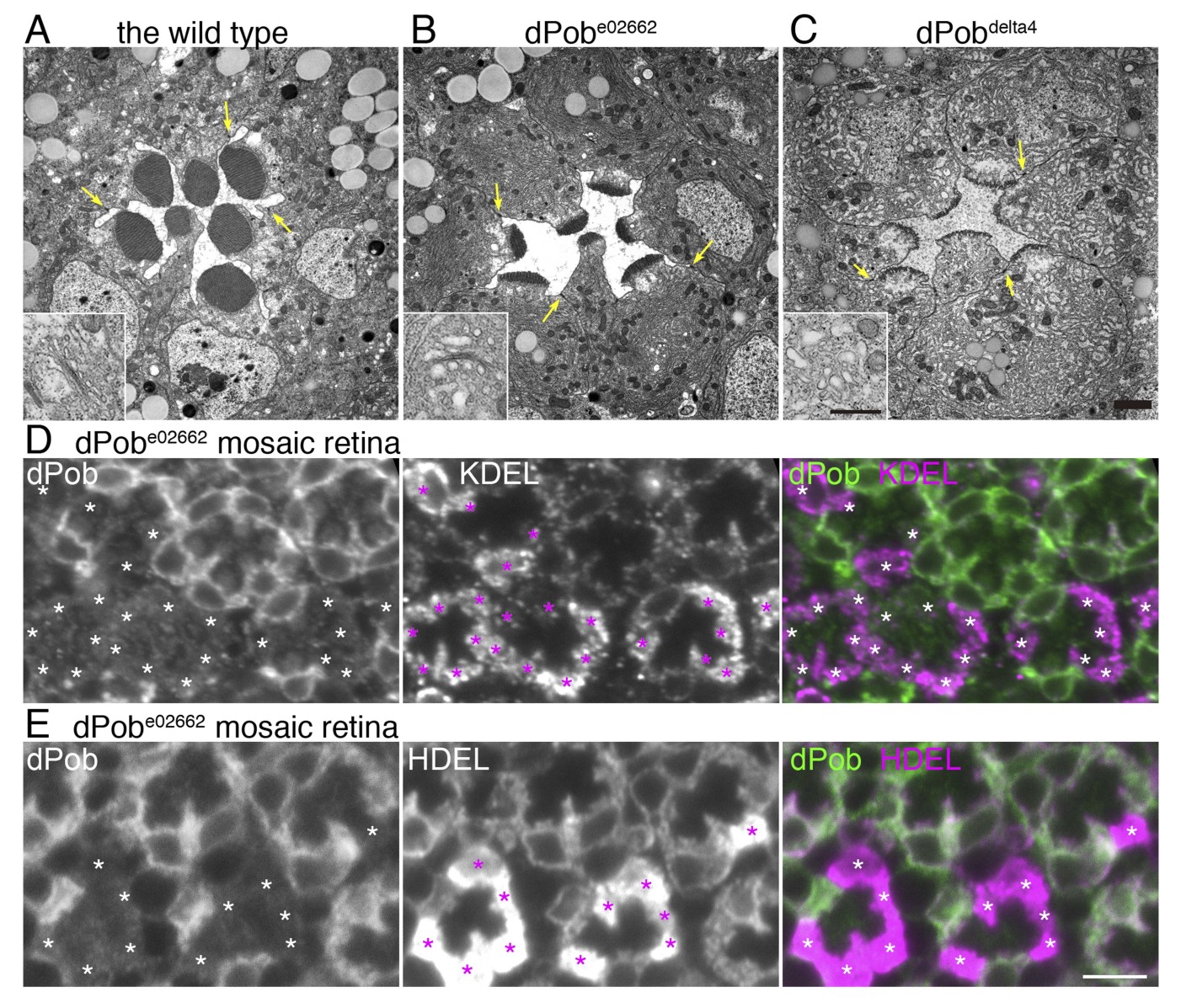

**Figure 8**. Endoplasmic reticulum membrane amplification and unfolded protein response (UPR) induced in *dPob*$^{\Delta 4}$ photoreceptor. (**A–C**) Electron microscopy of late pupal photoreceptors: *wild-type* (**A**), *dPob*$^{e02662}$ (**B**), and *dPob*$^{\Delta 4}$ photoreceptors (**C**). Arrow indicate adherens junctions. Insets show Golgi bodies. (**D**, **E**) Immunostaining of a *dPob*$^{e02662}$ mosaic retina. dPob is shown in green and KDEL (**D**) or HDEL (**E**) are shown in magenta. Asterisks show *dPob*$^{\Delta 4}$ homozygous photoreceptors. Scale bar: 1 µm (**A–C**), 5 µm (**D**, **E**).

Therefore, mutants lacking the function of a gene essential for folding or degradation of unfolded protein probably exhibit UPR. In fact, the yeast *Pob* homolog, *EMC3*, was identified by screening of mutants exhibiting upregulated UPR. ER amplification and chaperone induction, which we observed in *dPob*-deficient photoreceptors, are also common outcomes of UPR. We therefore examined whether UPR is induced in *dPob*-deficient photoreceptors. First we used the *Xbp1:GFP* sensor, which is an established method for detecting UPRs in flies (*Ryoo et al., 2007*). During UPR, Ire1 catalyzes an unconventional splicing of a small intron from the *xbp1* mRNA, enabling translation into an active transcription factor (*Yoshida et al., 2001*). Using this mechanism, *Xbp1:GFP* sensor, a fused transcript of *Drosophila* Xbp1 and GFP translated only after the unconventional splicing by Ire1, can be used as a reporter of one of the UPR transduction pathways (*Ryoo et al., 2007*). In both *dPob*$^{\Delta 4}$ and *dPob*$^{e02662}$

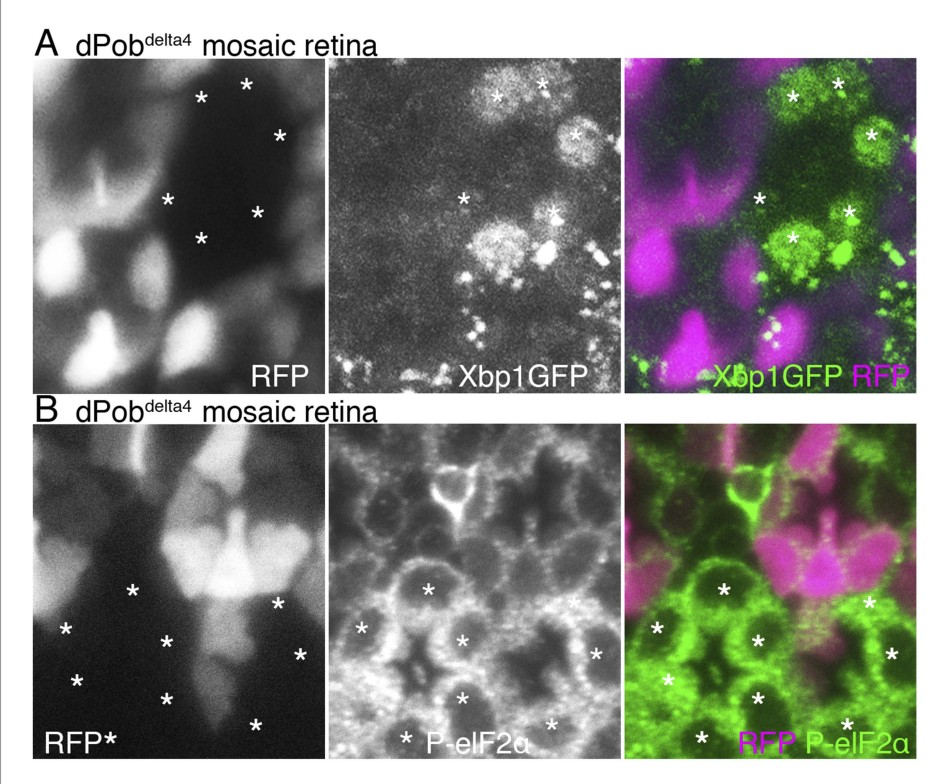

**Figure 9**. Unfolded protein response (UPR) induced in dPob$^{\Delta4}$ photoreceptor. (**A**) Projection image from the Z-series section with a 1 µm interval of dPob$^{\Delta4}$ mosaic retina expressing RFP (magenta) as a wild-type cell marker and Xbp1: GFP as a UPR sensor. The Xbp1:GFP signal (green) is enhanced by immunostaining using anti-GFP antibody. Asterisks show dPob$^{\Delta4}$ homozygous photoreceptors. (**B**) Immunostaining of a dPob$^{\Delta4}$ mosaic retina expressing RFP (magenta) as a wild-type cell marker. Phosphorylated eukaryotic translation Initiation Factor 2α is shown in green. Asterisks show dPob$^{\Delta4}$ homozygous photoreceptors.

mutant mosaic retinas expressed *Xbp1:GFP* sensor in all R1–6 photoreceptors, and *Xbp1:GFP* fusion proteins were detected in the dPob mutant photoreceptors but not in the wild-type (*Figure 9A* and data not shown). Next, we examined the level of eukaryotic translation Initiation Factor 2α (eIF2α) phosphorylation because UPR is well known to induce eIF2α phosphorylation to attenuate protein translation on the ER membrane in a transduction pathway independent from Irel/Xbp1 (*Ron and Walter, 2007*; *Cao and Kaufman, 2012*). Anti-phospho-eIF2α signals were stronger in both *dPob$^{\Delta4}$* and *dPob$^{e02662}$* photoreceptors than in wild-type photoreceptors (*Figure 9B* and data not shown). These results indicate that UPR is induced in the *dPob*-deficient photoreceptors, similar to *EMC* mutant.

## Rhabdomere development and degeneration in *dPob* null mutant

Because the synthesis of many membrane proteins was affected in dPob mutant cells, we observed the phenotype of dPob mutant throughout the developmental processes of photoreceptors. Despite the lack of many membrane proteins, ommatidial formation was not affected in *dPob$^{\Delta4}$* photoreceptors in mosaic retina; adherence junctions formed normally (*Figure 6E*) and the apical membrane was well differentiated into stalks and rhabdomeres (identified with Crb and phosphorylated moesin, respectively) (*Figure 6B* and data not shown) (*Karagiosis and Ready, 2004*). The IRS was formed normally and rhabdomeres were still separated by IRSs (*Figure 8A–C*). We observed *dPob$^{\Delta4}$* mosaic retinas at 58% and 75% pupal development (pd) by electron microscopy (*Figure 10A,B*). The wild-type photoreceptors at 58% pd had already begun to amplify the rhabdomere membranes. The rhabdomeres were shorter in *dPob$^{\Delta4}$* photoreceptors than in wild-

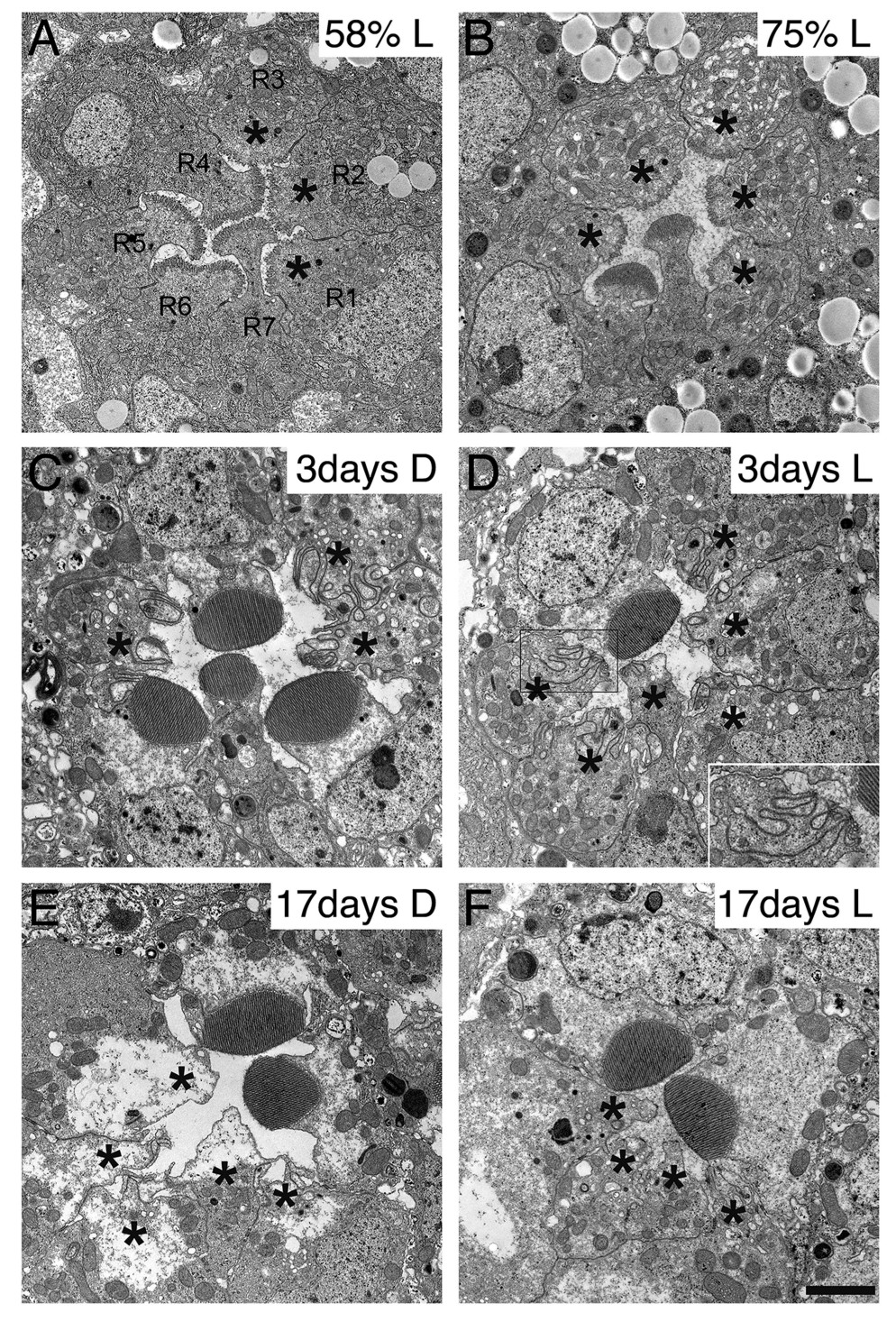

**Figure 10**. Development and degeneration of *dPob*[Δ4] photoreceptor rhabdomeres. Electron microscopy of pupal and adult *dPob*[Δ4] mosaic retinas. Asterisks show *dPob*[Δ4] homozygous photoreceptors. Scale bar: 1 μm. (**A**, **B**) *dPob*[Δ4] mosaic ommatidia from 58% pupal development (**A**) and 73% pupal development (**B**) under constant light (L) condition. (**C–F**) *dPob*[Δ4] mosaic ommatidia from flies reared in complete darkness (D) (**C**, **E**) or under 12 hr light/12 hr dark conditions (**D**, **F**). Ommatidia from 3-day-old (**C**, **D**) and 17-day-old (**E**, **F**) flies. (**D**, inset) *dPob*[Δ4] R5 photoreceptor rhabdomere at higher magnification.

type photoreceptors, but the difference in their appearance was subtle at this stage. Until 75% pd, the microvilli of wild-type rhabdomeres were ~0.5 µm long and packed tightly. However, the microvilli of $dPob^{\Delta 4}$ rhabdomeres at 73% pd retained almost the same length and appearance as those at 58% pd, which is the same as the $dPob^{\Delta 4}$ rhabdomeres of the late pupal retina (*Figures 10A,B and 8C*). ER membrane expansion and dilation were already apparent at 58% pd. These results indicate that dPob does not inhibit overall photoreceptor development and morphogenesis but does affect microvilli elongation and rhabdomere formation.

Because zebrafish *pob* was identified as the responsible gene of $pob^{a1}$ mutant which exhibits red cone photoreceptor degeneration (*Brockerhoff et al., 1997*; *Taylor et al., 2005*), we investigated photoreceptor degeneration of the *dPob* null mutant. Three-day-old $dPob^{\Delta 4}$ mosaic retinas from flies reared under dark or 12 hr light/12 hr dark cycles were observed by electron microscopy (*Figure 10C, D*). In both conditions the rhabdomeres of $dPob^{\Delta 4}$ photoreceptors invaginated into the cytoplasm, indicating that *dPob*-deficient rhabdomeres undergo retinal degeneration in a light-independent manner, like Rh1 null mutants (*Kumar and Ready, 1995*). No microvilli or invaginations were observed in 17-day-old $dPob^{\Delta 4}$ mosaic retinas, suggesting most invaginated microvilli had degraded before day 17 (*Figure 10E,F*). Such rhabdomere degeneration was observed not only in R1–6 peripheral photoreceptors but also in R7 central photoreceptors. Therefore, dPob is an essential protein for maintenance of retinal structure, similar to the zebrafish *pob* gene.

## Discussion

The present study shows that dPob, the *Drosophila* homolog of a subunit of EMC, EMC3, localizes in the ER and is essential for Rh1 accumulation of the rhabdomeres. The deficiency of each of two other EMC subunits, EMC1 and EMC8/9, also shows absence of Rh1 on the rhabdomeres. Mammalian EMC8 and EMC9 were identified together with EMC7 and EMC10 by high-content proteomics strategy (*Christianson et al., 2011*). Unlike EMC1–6 subunits, EMC8 and EMC9 do not have a transmembrane helix or signal peptide and no experimental data have been reported to show the functions of these subunits. We observed that *Drosophila* EMC8/9-deficient cells lack accumulation of Rh1 apoprotein in the ER and impaired biosynthesis of the multi-pass transmembrane proteins. These phenotypes in EMC8/9 deficiency are indistinguishable from those in dPob and EMC1 mutant cells, suggesting that EMC8/9 work together with EMC1 and dPob. This is the first functional study of the additional subunits of EMC, which are lacking in yeast.

We found that null mutants of EMC subunits are defective in expressing the multi-pass transmembrane proteins rhodopsins, TRP, and the alpha subunit of $Na^+K^+$-ATPase, which have seven, six, and eight transmembrane helices, respectively. In contrast, the EMC null mutants adequately express type I, type II, or type IV single-pass membrane proteins. Our observation on the substrate specificity of EMC is mostly consistent with previous reports. *Jonikas et al. (2009)* found that EMC mutants and a strain overexpressing a misfolded transmembrane protein, sec61-2p or KWS, had a similar genetic interaction pattern and suggested that EMC works as a chaperone for transmembrane proteins. A recent study in *Caenorhabditis elegans* using a hypomorphic EMC6 allele and RNAi knock-down of emc1–6 genes showed results partially consistent with our study; at least two pentameric Cys-loop receptors, AcR and $GABA_A$, consisting of subunits with four transmembrane helices, were significantly decreased in the hypomorphic EMC6 mutants but GLR-1, a tetrameric AMPA-like glutamate receptor with four transmembrane helices and a type I single-pass trans-membrane EGF receptor, was not affected (*Richard et al., 2013*). Despite its four transmembrane helices, GLR-1 was normally expressed in the hypomorphic emc6 mutant of the nematode; however, these results may indicate that the residual activity of EMC was sufficient for the expression of GLR-1. The degree of requirement of EMC activity can vary for each membrane protein. In fact, in a dPob hypomorphic allele, $dPob^{e02662}$, near-normal expression of $Na^+K^+$-ATPase was detected (*Figure 6I*) despite a severe reduction in a dPob null allele, $dPob^{\Delta 4}$. Overall, the results observed in the dPob null mutant does not conflict with previous studies but rather clarifies the role of EMC in the biosynthesis of multi-pass transmembrane proteins. Because of the limited availability of antibodies, we could not show a clear threshold for the number of transmembrane helices in the substrates for EMC activity. In total, the data presented to date indicate that EMC affects the expression of membrane proteins with four or more transmembrane helices.

Co-immunoprecipitation of dPob/EMC3 and Cnx by EMC1 indicates that EMC components and Cnx can form a complex. The photoreceptors of an amorphic mutant of Cnx show complete loss of

Rh1 apoprotein (*Rosenbaum et al., 2006*), just as shown in dPob, EMC1 or EMC8/9 mutants. Moreover, both Cnx and EMC3 are epistatic to the mutant of the rhodopsin-specific chaperone, NinaA, which accumulates Rh1 apoprotein in the ER. These results indicate that EMC and Cnx can work together in the Rh1 biosynthetic cascade prior to NinaA. Cnx, the most studied chaperone of N-glycosylated membrane proteins, recognizes improperly folded proteins and facilitates folding and quality control of glycoproteins through the calnexin cycle, which prevents ER export of misfolded proteins (*Williams, 2006*). One possible explanation for our result is that the EMC-Cnx complex is required for multi-pass membrane proteins to be incorporated into the calnexin cycle. If the EMC-Cnx complex is a chaperone of Rh1, physical interaction is expected between ER-accumulated Rh1 apoprotein and the EMC-Cnx complex. Indeed, it is reported that Cnx is co-immunoprecipitated with *Drosophila* Rh1 (*Rosenbaum et al., 2006*). However, in this study, Rh1 apoprotein accumulated in the chromophore-depleted photoreceptor cells was not co-immunoprecipitated with EMC1. Thus, even if EMC is a Rh1 chaperone, our result indicates that EMC is unlikely to be working in the calnexin cycle or acting as a buffer of properly folded Rh1 apoprotein ready to bind the chromophore 11-*cis* retinal.

In addition to preventing the export of immature protein by the calnexin cycle, Cnx is also known to recognize the nascent polypeptides co-translationally (*Chen et al., 1995*). The dual role of Cnx might explain the observations that both dPob/EMC3 and Cnx are epistatic to another ER resident chaperone, NinaA, whereas Cnx but not the EMC-Cnx complex binds to Rh1. These results imply that the EMC-Cnx complex is more likely to be involved in the earlier processes such as membrane integration or co-translational folding than in the folding of fully translated membrane-integrated Rh1 apoprotein.

In spite of the absence of Rh1 apoprotein, UPR is much more upregulated in the EMC3 null mutant than in the NinaA null mutant which accumulates Rh1 apoprotein in the ER. The elevated UPR without accumulation of Rh1 apoprotein in the dPob mutant photoreceptor can be explained either by the quick degradation of Rh1 apoprotein or by accumulation of the single-pass membrane proteins abandoned by the multi-pass binding partner.

Newly synthesized secreted proteins co-translationally translocate across the membrane through the translocons Sec61 in eukaryotic ER or SecYEG in the plasma membrane of bacteria. The translocons also mediate integration of the transmembrane helix of the integral membrane protein into the lipid bilayer (*Park and Rapoport, 2012*). In bacteria, mitochondria and chloroplasts, YidC/Oxa1/Alb3 proteins specifically facilitate insertion, folding, and assembly of many transmembrane proteins (*Wang and Dalbey, 2011*). In the ER membrane of eukaryotes, in addition to the translocon, other components such as translocon-associated protein/signal sequence receptor (TRAP/SSR) complex and translocating chain-associating membrane protein (TRAM) complex are required for the membrane insertion of the transmembrane helix. Most of the newly synthesized multi-pass membrane proteins are co-translationally integrated into the ER membrane through the translocon complex. Although the mechanism of this process is yet to be fully understood, it is assumed that only one or two transmembrane helices can be stored in the translocon channel and the lateral gate and that the next set of newly synthesized transmembrane helices displace them (*Rapoport et al., 2004*; *Cymer et al., 2014*). In the case of nascent chain of bovine rhodopsin, translocon associates with transmembrane helices sequentially, and TRAM temporarily associates with the second transmembrane helix (*Ismail et al., 2008*). EMC may be involved in these co-translational membrane integration or co-translational folding processes.

Zebrafish *pob* was identified as the responsible gene of *pobᵃ¹* mutant, which exhibits red cone photoreceptor degeneration (*Brockerhoff et al., 1997*; *Taylor et al., 2005*). Because only red cone photoreceptors degenerated in zebrafish *pobᵃ¹* mutant, *pob* is postulated as a gene with a red cone-specific function. However, the identification of the *pobᵃ¹* mutation as hypomorphic together with *pob* expression in all photoreceptors, as well as its localization in the early secretory pathway, suggests that Pob has a general function rather than being red cone-specific (*Taylor et al., 2005*). We found that *dPob*-deficient rhabdomeres undergo retinal degeneration in a light-independent manner, like Rh1 null mutants (*Kumar and Ready, 1995*). Rhabdomere degeneration was observed not only in R1–6 peripheral photoreceptors but also in R7 central photoreceptors. Our results indicate that dPob is an essential protein for the maintenance of retinal structure, similar to the zebrafish *pob* gene.

# Materials and methods

### *Drosophila* stocks and genetics

Flies were reared at 20–25°C in 12 hr light/12 hr dark cycles and fed standard cornmeal/glucose/agar/yeast food unless noted otherwise. Vitamin A-deficient food contained 1% agar, 10% dry yeast, 10% sucrose, 0.02% cholesterol, 0.5% propionate, and 0.05% methyl 4-hydroxybenzoate.

UAS-Xbp1::GFP was a gift from H Ryoo at New York University and other *Drosophila* stocks obtained from Bloomington Stock Center (BL) or the Kyoto Drosophila Genetic Resource Center (KY) are referred to with their respective sources and stock numbers.

dPob deletion mutants were made using a standard induced FLP/FRT recombination method (*Parks et al., 2004*). Trans-heterozygous $PBac(WH)^{f07762}$ (BL19109) and $P (RS3)^{CB-0279-3}$ (KY123106) males carrying hs-FLP (BL6876) were heat treated three times at 37°C for 1 hr at larval stages. SM6a-balanced offspring were genotyped using PCR to select the recombinant carrying both the proximal side of $PBac(WH)^{f07762}$ and the distal side of $P (RS3)^{CB-0279-3}$ with the following primers: 5′-CTCCTTGCCAGCTTCTGC-3′ and 5′-TCGCTGTCTCACTCAGACTCA-3′ for $P (RS3)^{CB-0279-3}$, and 5′–CCACCGAAGAGGCCTACTATT-3′ and 5′-TCCAAGCGGCGACTGAGATG-3′ for $PBac(WH)^{f07762}$.

### Transgenic flies for UAS-dPob, UAS-EMC1::GFP

The entire coding region of the dPob gene was amplified from a cDNA clone LD37839 (DGRC: Drosophila Genomics Resource Center, Bloomington, IN, USA) and cloned into pTW (DGRC) to construct pP{UAST-dPob}. To construct pP{UAST-EMC1::GFP}, the entire coding region of CG2943 except the stop codon was amplified from a cDNA clone LD19064 (DGRC) and cloned into pTWG (DGRC). Plasmids were injected into embryos by BestGene Inc. (Chino Hills, CA, USA) to generate transgenic lines.

### Live imaging of fluorescent proteins expressed in photoreceptors

Fluorescent proteins expressed in photoreceptors were imaged by water-immersion technique.

y w ey-FLP;CG6750$^{e02662}$ FRT40A/ CyO y+ (KY114504) was mated with w;P3RFP FRT40A/SM1;Rh1-Arrestin2::GFP eye-FLP/TM6B (*Satoh et al., 2013*). Late pupae of the siblings with GFP-positive RFP mosaic retina were attached to the slide glass using double-sided sticky tape and the pupal cases around the heads were removed. The pupae were chilled on ice, embedded in 0.5% agarose, and observed using an FV1000 confocal microscope equipped with a LUMPlanFI water-immersion 40× objective (Olympus, Tokyo, Japan). Arrestin2::GFP specifically binds to activated rhodopsin (*Satoh et al., 2010*). Rh1 was activated by a 477 nm solid-state laser to bind Arr2:GFP and GFP. The wild-type marker P3RFP is DsRed gene under the control of three Pax3 binding sites and labels photoreceptors (*Bischof et al., 2007*).

### EMS mutagenesis and screening

The precise method of screening, whole genome re-sequencing, will be described elsewhere. Briefly, second or third chromosomes carrying P-element vector with FRT on 40A, 42D, or 82B (*Berger et al., 2001*) were isogenized and used as the starter strains. EMS was fed to males in a basic protocol (*Bökel, 2008*) and mosaic retinas were generated on F1 or F2. The estimated number of lethal mutations introduced per chromosome arm was 0.8–1.8. The mutants were screened based on the distribution of Arr2-GFP by confocal live imaging under water-immersion lens using 3xP3-RFP as the wild-type marker, as previously described for the screening of insertional mutants (*Satoh et al., 2013*).

### Mapping and determination of mutations

Meiotic recombination mapping was carried out by the standard method (*Bökel, 2008*). Briefly, to allow meiotic recombination between the proximal FRT, the phenotype-responsible mutation and a distal miniature w+ marker, flies carrying isogenized chromosome of 008J and 655G were crossed with flies with isogenized P{EP755} and P{EP381} which carry miniature-w+ marker, respectively. Female offspring carrying the mutated chromosome and the miniature-w+-marked chromosome were crossed with males carrying FRT42D, P3RFP, and Rh1Arr2GFP. The resulting adult offspring with w+ mosaic, which means maternally inherited both FRT and w+, were observed using live imaging to judge whether the mutation responsible for the dPob-like phenotype had been

inherited. The recovered flies were individually digested in 50 µl of 200 ng/µl Proteinase K in 10 mM Tris-Cl (pH 8.2), 1 mM EDTA, and 25 mM NaCl at 55℃ for 1 hr and heat inactivated at 85℃ for 30 min and at 95℃ for 5 min. 0.5 µl of the digested solution were used as the template of PCR amplification for RFLP analysis according to the method described in the FlySNP database (*Chen et al., 2008*; http://flysnp.imp.ac.at/index.php). The mutation responsible for the dPob-like phenotype of 008J was mapped between SNP markers 1417 and 1518 defined in the FlySNP database.

## Whole-genome and targeted re-sequence of EMS-generated mutants

For the whole genome re-sequencing of the 008J mutant, the second chromosome was balanced over a balancer, CyO, P{Dfd-GMR-nvYFP}(Bloomington stock number 23230) to facilitate the isolation of homozygous embryo. Using REPLI-G single cell kit (QIAGEN, Hilden, Germany), the genomic DNA was amplified from two 008J homozygous embryos independently. A sequencing library was prepared using Nextera DNA sample preparation kit (Illumina, San Diego, CA, USA) for each embryo and 2 × 250 bp reads were obtained using MiSeq v2 kit (Illumina). Reads were mapped to release five of the *Drosophila melanogaster* genome using BWA 0.7.5a. The RFLP-mapped region of 008J was covered by reads with an average depth of 23.2× and width of 99.5%. Mapped reads were processed using picard-tools 1.99 and Genome Analysis Tool Kit 2.7-2 (GATK, Broad Institute, Cambridge, MA, USA). SNVs and Indels were called using Haplotypecaller in GATK. SNVs and Indels were subtracted by the ones of the isogenized starter stock to extract the unique variants in 008J and annotated using SnpSift (*Cingolani, 2012*). The point mutation on 2R:18770005 was verified by capillary sequencing of PCR-amplified fragment using 5′ GTCGCGGTCACACTTTCTAG 3′ and 5′ CTGCAGCGTCATCAGTTTGT 3′ as primers.

For targeted re-sequencing of 655G, a region including CG2943 was amplified from a heterozygous fly of the 655G mutant chromosome and the starter chromosome using KOD FX Neo DNA polymerase and 5′ TTTTGTTCTTGTTGGGCGACTCCTTTTCCGTCTC 3′ and 5′ AGGCTGTGTCTTTGTTGTTTTGGCGTTGTCGTC 3′ as primers. Reads covering the CG2943 gene region at a depth of 2213–6436 were obtained using MiSeq and mapped, as described above. The sequence was confirmed by capillary sequencing and PCR using 5′ GCAAGAATCC CATCGAGCAT 3′ and 5′ CCTTCTTCACGTCCCTGAGT 3′ as primers.

## Antisera against dPob and CNX99a

Fragments of cDNA encoding V28-D104 (dPob-N) or G173-S247 (dPob-C1) of *dPob* were amplified from a cDNA clone, LD37839 (Drosophila Genomics Resource Center, Bloomington, IN, USA) and cloned into *pDONR-211* using Gateway BP Clonase II and then into *pET-161* expression vector using Gateway LR Clonase II (Life Technologies, Carlsbad, CA, USA). The fusion proteins with 6xHis-tag were expressed in BL21-Star (DE3) (Life Technologies) and purified using Ni-NTA Agarose (QIAGEN). To obtain antisera, rabbits were immunized six times with 300 µg dPob-N fusion protein (Operon, Tokyo, Japan) and three rats were immunized six times with 125 µg dPob-C1 fusion protein (Biogate, Gifu, Japan). Antisera against *Drosophila* Cnx were raised by immunizing a rabbit four times with 400 to 200 µg of synthetic peptide corresponding to C-terminal 24 amino acids of Cnx99a protein conjugated to KLH (Sigma Aldrich Japan, Tokyo, Japan).

## Immunoblotting

Immunoblotting was performed as described previously (*Satoh et al., 1997*). The antibodies used were as follows: rabbit anti-dPob–N-terminal (dPob-N) (1:2000 concentrated supernatant) (made by the authors of this paper), three rat anti-dPob–C-terminal antibodies (dPob-C1-3) (1:2000 concentrated supernatant) (made by the authors of this paper) as primary antibodies. HRP-conjugated anti-rat or anti-rabbit IgG antibody (1:20,000, Life Technologies) was used as a secondary antibody. For co-immunoprecipitation, 1: 2000 rabbit anti-dPob-N, 1:2000 rabbit anti-Cnx99A, 1:2000 rabbit anti-GFP (Life Technologies), mouse anti-Rh1 monoclonal antibody 4C5, and detected by biotinylated secondary antibodies followed by HRP-conjugated avidin. Signals were visualized using enhanced chemiluminescence (Clality Western blotting ECL Substrate; BioRad, Hercules, CA, USA) and imaged using ChemiDoc XRS+ (BioRad).

## Immunohistochemistry

Fixation and staining were performed as described previously (*Satoh and Ready, 2005*). The primary antisera were as follows: rabbit anti-Rh1 (1:1000) (*Satoh et al., 2005*), chicken anti-Rh1 (1:1000)

(*Satoh et al., 2013*), mouse monoclonal anti-HDEL (1:100) (Santa Cruz Biotechnology, Dallas, TX, USA), mouse monoclonal anti-KDEL (1:100) (Assay Designs, Ann Arbor, MI, USA), rabbit anti-NinaA (1:300) (gift from Dr Zuker, Colombia University), mouse monoclonal anti-Na$^+$K$^+$-ATPase α subunit (1:500 ascite) (DSHB, Iowa City, IA, USA), rat monoclonal anti-DE-cad (1:20 supernatant) (DSHB), mouse monoclonal anti-Syx1A (1:20 supernatant) (DSHB), mouse monoclonal anti-Nrt (1:20 supernatant) (DSHB), mouse monoclonal anti-Nrv (1:20 supernatant) (DSHB), mouse monoclonal anti-FasIII (1:20 supernatant) (DSHB), mouse monoclonal anti-Nrg (1:20 supernatant) (DSHB), mouse monoclonal anti-Chp (24B10) (1:20 supernatant) (DSHB), rat anti-Crb (gift from Dr Tepass, University of Toronto), rabbit anti-TRP (gift from Dr Montell, Johns Hopkins University), rabbit anti-dMPPE (1:50) (gift from Dr Han, Southeast University), and rabbit anti-phosphorylated eIF2α (1:300) (Cell Signaling Technologies, Danvers, MA, USA). The secondary antibodies used were anti-mouse, rabbit, rat, and chicken IgG labeled with Alexa Fluor 488, 568, and 647 (1:300) (Life Technologies) and Cy2 (1: 300) (GE Healthcare Life Sciences, Pittsburgh, PA, USA). Samples were examined and images recorded using a FV1000 confocal microscope (60×, 1.42-NA lens; Olympus, Tokyo, Japan). To minimize bleed-through, each signal in double- or triple-stained samples was imaged sequentially. Images were processed in accordance with the guidelines for proper digital image handling using ImageJ and/or Adobe Photoshop CS3.

## Co-immunoprecipitation analysis of EMC complex

The EMC1 gene was cloned into a P-element vector pTWG using the Gateway System (Life Technologies) to express EMC1 protein-tagged GFP on the C-terminus under control of upstream activation sequence (UAS). Transgenic lines were generated by the BestGene Inc. (Chino Hills, CA, USA). UAST-EMC1-GFP(1M), a line carrying the transgene on the second chromosome, was crossed to Rh1-Gal4 line to express EMC1-GFP in the photoreceptor or to hs-Gal4 line to express EMC1-GFP in the whole body. A protein-trap line, Sec61alpha [ZCL0488] which constitutively expresses GFP-tagged Sec61alpha protein, was used as a control. To accumulate rhodopsin in the ER, flies were reared in the vitamin A-deficient medium in a Rh1-driven experiment. For heat-shock driven expression, newly eclosed adult fly flies were incubated at 37°C for 45 min a day before preparation. Within 0–1 days after eclosion, flies were frozen with liquid nitrogen and stored at −80°C. The heads were collected by sieving in liquid nitrogen, ground to powder and homogenized in buffer (50 mM Tris-Cl, 500 mM NaCl, pH 7.5) containing 1:200 Protein inhibitor cocktail VI (Calbiochem, San Diego, CA, UAS) using BioMasher II (Wako Pure Chemical, Osaka, Japan) with motor drive. Debris was removed by centrifugation at 950×g for 5 min and the membrane was precipitated by centrifugation at 21,500×g for 15 min. Approximately 30 μl of membrane pellet were solubilized by 130 μl of 1% CHAPS and placed on ice for 1 hr, and the insoluble membrane was removed by centrifugation at 21,500×g for 30 min. The extract was diluted fivefold by the buffer and 50 μl of Anti-GFP-Magnetic beads (MBL, Nagoya, Japan) were added and mixed by mild rotation for 18 hr. The magnetic beads were rinsed with 2× 100 μl of 0.1% CHAPS in buffer and the bound protein was extracted by incubation in 20 μl SDS-PAGE Sampling Buffer (BioRad) for 5 min at room temperature and an equal amount of Sampling Buffer with 2-mercaptoethanol was then added. The extracts were heat denatured for 5 min at 37°C. SDS-PAGE and immunoblotting was performed as described above.

## Electron microscopy

Electron microscopy was performed as described previously (*Satoh et al., 1997*). Samples were observed on a JEM1200 or JEM1400 electron microscope (JEOL, Tokyo, Japan).

## Quantification of relative expression of mRNA of Rh1, TRP, and Arr2 normalized by Act5C

Whole-eye mutant clones were generated using the FRT/GMR-hid method (*Stowers and Schwarz, 1999*). Both eyes were dissected from two adult flies per sample and cDNA was reverse-transcribed using SuperPrep Cell Lysis and RT Kit for qPCR (Toyobo, Osaka, Japan) according to the manufacturer's instructions. Eyes with whole-eye clones of FRT40A were used as a control to obtain the relative standard curves. qPCR reactions were performed using the StepOne real-time PCR system (Life Technologies) and KOD SYBR qPCR Mix (Toyobo, Osaka, Japan), according to the manufacturers' instructions. PCR condition was 98°C for 2 min, followed by 40 cycles at 98°C for 15 s, 55°C for 15 s, and 68°C for 45 s, and a melt curve stage of 95°C for 30 s, 60°C for 1 min, and 0.3°C/s increments to 98°C,

with primers of Rh1: (ninaE-qF1:5′-GTGGACACCATACCTGGTC-3′ and ninaE-qR1:5′-GCGA-TATTTCGGATGGCTG-3′), Arr2: (Arr2-qF1:5′-AAGGATCGCCATGGTATCG-3′ and Arr2-qR1:5′-TACGAGATGACAATACCACAGG-3′), TRP: (Trp-qF2:5′-GAATACACGGAGATGCGTC-3′ and Trp-qF2:5′-CTCGAGTTCCATGGATGTG-3′), Act5C: (5′-GCTTGTCTGGGCAAGAGGAT-3′ and 5′-CTGGAACCACACAACATGCG-3′). The relative expression levels were normalized by Act5C.

## Acknowledgements

We thank Drs U Tepass, C Montell, C Zuker, H Ryoo, and J Han who kindly provided fly stocks and reagents. We also thank the Bloomington Stock Center and the *Drosophila* Genetic Resource Center of the Kyoto Institute of Technology for fly stocks. This study was supported by grants from the Naito Foundation (25-040920), the Novartis Foundation (25-050421), the Hayashi Memorial Foundation for Female Natural Scientists (25-051022), PRESTO (25-J-J4215), and KAKENHI (21687005, 21113510, and 23113712) to ASK. This study was also supported by grants from the Global Centers of Excellence Program 'Advanced Systems-Biology: Designing The Biological Function' from the Japanese Ministry of Education, Culture, Sports, Science, and Technology. Whole genome and targeted re-sequencing was carried out at the Analysis Center of Life Science, Natural Science Center for Basic Research and Development, Hiroshima University.

## Additional information

### Funding

| Funder | Grant reference number | Author |
| --- | --- | --- |
| Japan Science and Technology Agency (JST) | PRESTO (Precursory Research for Embryonic Science and Technology) 25-J-J4215 | Akiko K Satoh |
| Japan Society for the Promotion of Science (JSPS) | 21687005 KAKENHI | Akiko K Satoh |
| Japan Society for the Promotion of Science (JSPS) | 21113510 KAKENHI | Akiko K Satoh |
| Japan Society for the Promotion of Science (JSPS) | 30529037 KAKENHI | Akiko K Satoh |
| The Naito Foundation | 25-040920 | Akiko K Satoh |
| The Novartis Foundation | 25-050421 | Akiko K Satoh |
| The Hayashi Memorial Foundation for Female Natural Scientists | 25-051022 | Akiko K Satoh |

The funders had no role in study design, data collection and interpretation, or the decision to submit the work for publication.

### Author contributions

TS, AKS, Conception and design, Acquisition of data, Analysis and interpretation of data, Drafting or revising the article; AO, ZL, TI, Acquisition of data, Drafting or revising the article

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
