## [Decision Letter]

Congratulations: we are very pleased to inform you that your article, “dPob/EMC is essential for biosynthesis of rhodopsin and other multi-pass membrane proteins in *Drosophila* photoreceptors”, has been accepted for publication in *eLife*, subject to revisions of grammar and text. We also hope you would give due consideration to the minor comments of reviewers 1 and 3. The Reviewing Editor for your submission was David Ron.

Reviewer #1

In Figure 5, right panel it is not clear why the sec61::GFP input shows a band the size of the EMC1::GFP; what is the nature of this band?

Equally why is the +/- vitamin A experiment performed with different driver lines? wouldn't it be better to do both with the Rh1-Gal4 driver? Please give the rationale of using the two drivers.

Reviewer #3

1) The authors show that EMC1::GFP does not co-immuno-precipitate with Rh1. Does calnexin co-IP with Rh1? I ask because the authors conclude (in the Discussion) that “ EMC is unlikely to be working in the calnexin cycle”, and it is difficult for me to understand why the authors have come to that conclusion.

2) The authors report negative results with the ERAD component mutants, EDEM1, EDEM2 and VCP/+. I don't see the point of showing such negative data, as one cannot draw any conclusions. Perhaps the individual EDEM mutants do not show Rh1 phenotype as they are redundant (in mammals, EDEMs are known to be genetically redundant). Also, VCP's effect was assessed in a heterozygous condition, and a negative result here simply means that there is no dominant genetic effect. How about analyzing homozygous clones? Alternatively, since ERAD is not a major point of this work, the authors may want to take out this part, as it does not contribute to the overall story.

[Editors’ note: a previous version of this study was rejected after peer review, but the authors submitted for reconsideration. The previous decision letter after peer review is shown below.]

Thank you for choosing to send your work entitled “dPob is essential for rhodopsin maturation in *Drosophila* photoreceptors” for consideration at *eLife*. Your full submission has been evaluated by Randy Schekman (Senior editor) and 3 peer reviewers, one of whom is a member of our Board of Reviewing Editors. Between them, the three reviewers had expertise in *Drosophila* genetics, the study of assembly and maturation of membrane transporters and ER protein folding homeostasis. Their decision was reached after discussions between the reviewers.

The reviewers noted the significance of your discovery of a role for a *Drosophila* member of the EMC complex in the maturation of rhodopsin. Your discovery that dPob/EMC3 acts upstream of ninaA in rhodopsin maturation was specifically credited with supporting a role for this member of the EMC complex in the early steps of the maturation of a multi-pass membrane protein. However, the expert reviewers were unanimous in their view that these finding do not add up to an advance of sufficient measure to merit publication in *eLife*.

It is possible that a different manuscript, for example one with further details on genetic lesions in other *Drosophila* EMC homologs or one that carried the biochemical analysis of the dPob/EMC3 lesion further would had received a warmer welcome at *eLife*. Unfortunately, the manuscript before us is too far off this mark to suggest specific experiments that would render it suitable for this journal.

Thus we are left with no choice but to return the manuscript to you in the hope that you will find the reviewers’ comments (appended below) of use.

Reviewer #1

A mosaic screen for mutations in (otherwise essential) genes required for rhodopsin trafficking and maturation in the fly eye led the authors to dPod, in whom homozygosity for a hypomorphic allele compromised Rh1 expression in the rhabdomeres.

dPod is a fly homolog of a zebra fish gene mutations in which compromise color vision and of EMC3 a yeast gene whose compromise activates the unfolded protein response and whose encoded protein forms a stable complex with several other proteins that collectively constitute the ER Membrane Complex (or EMC).

Since its identification in 2009, the EMC has been postulated to play a role in the biogenesis of transmembrane proteins. This paper provides several pieces of information that support that notion.

The most important findings pertain to:

1) The failure of Rh1 to accumulate in the ER of dPod mutant flies that are deprived of vitamin A.

2) The epistatic relationship between dPod and ninaA, wherein the lack of dPod preempts the accumulation of Rh1 normally obsereved in ninaA mutants.

3) The selectivity of dPod mutations in compromising the trafficking of multi-pass transmembrane proteins, whilst preserving that of secreted and single pass proteins.

Together these findings point to site of dPod action early during the biogenesis of multi-pass transmembrane proteins.

Other findings, such as the altered ER morphology in mutant tissue and the activation of the UPR are less informative and more anticipated by the yeast work.

Thus the crucial issue for the reviewers is to establish the strength of the experimental data and, importantly the degree to which it advances our understanding of the EMC's biological role. If the application of this *Drosophila* genetic system has provided strong evidence favoring a selective role for this EMC component in early biogenesis of multi-pass transmembrane proteins, then the paper would be of interest to a broad readership regardless of the degree to which these findings were anticipated by prior speculation. If, however, the additional experimental data derived from the *Drosophila* system were deemed merely incremental, the paper would be more suited for a specialist journal.

Reviewer #2

In this manuscript, Satoh and colleagues report a genetic analysis of dPob, a *Drosophila* homolog of zebrafish Pob and yeast EMC4. The data presented here supports the idea that this gene is involved in the maturation of rhodopsins and other multipass transmembrane domain proteins in the endoplasmic reticulum. Consistently, the loss of dPob leads to the activation of the Unfolded Protein Response, and a defect in rhabdomere development.

EMC genes have been identified through a large-scale yeast genetic interaction screen, but their precise physiological roles had remained unclear. A couple of reports, based on studies in zebrafish and *C. elegans*, indicate that they are involved in membrane protein maturation. Although the authors have done a solid job in their characterization of dPob function in *Drosophila*, it appears to be an extension of the *C. elegans* work (28), but with more analysis of potential substrates. Moreover, this study does not address many of the pressing questions regarding the EMC complex. Does dPob's role in *Drosophila* reflect the entire EMC complex function? What is the mechanistic basis of the specificity of dPob towards multispan membrane proteins? Based on these grounds, whether the overall novelty and scope of this study justifies the publication of *eLife* can be subject to debate. Below are a few specific comments along these lines.

1) The authors had identified dPob through a previously reported genetic screen (38). Were there other EMC homologs identified in the screen? If it is the case, the authors should highlight those results, as it would indicate that the observed effect is not an isolated function of dPob.

2) EMC genes are grouped because of their genetic interactions with each other in yeast. This study is potentially significant because the function of EMCs still remain poorly understood. Naturally, one must ask whether other EMC gene homologs show similar effects in rhodopsin maturation or not. The authors can examine this through in vivo RNAi, or better, by use of available mutant alleles.

3) Does dPob physically interact with rhodopsin and other multipass membrane proteins? Immuno-precipitation experiments may be useful.

4) Is there any mechanistic insight as to how dPob would specifically recognize multipass membrane proteins? Although the authors show the dPob locus in Figure 1, they do not describe any domain structures in the gene. The fact that dPob specifically affects transmembrane domain is an intriguing phenomenon, but a mechanistic insight is lacking.

5) Does dPob physically interact with rhodopsins and other targets? A simple immune-precipitation experiment should provide answers.

Reviewer #3

The manuscript by Satoh et al. adds dPob, a subunit of the *Drosophila* EMC complex, to the list of factors required for rhodopsin biogenesis, an important process with biomedical relevance. The authors show that in the absence of dPob several rhodopsins do not mature and their steady-state levels are strongly decreased. Besides rhodopsins, two other polytopic membrane proteins, Na, K-ATPase alpha subunit and the TRP channel, were affected while four proteins with a single transmembrane segment and one secreted protein were not affected. These are interesting observations that do indeed suggest a role of dPob in the early biogenesis of polytopic membrane proteins.

However, I have major doubts whether the set of experiments adds much to our understanding of the specific function of dPob. Chaperone activity and protein folding are mentioned all over the manuscript but neither are ever addressed directly. Instead, steady-state levels of proteins and induction of the UPR are used as proxies to infer the folding status of the putative substrates. Both phenomena could be rather indirectly caused by lack of dPob function. In my opinion the manuscript does not achieve more than narrowing down which proteins are affected by loss of dPob function in a non-systematic fashion. The numbers of substrates in the affected and unaffected classes are low (furthermore, 'with a single membrane-spanning domain' is not an accurate topological classification for a membrane protein). Potentially, the distinctions between affected and unaffected proteins could fall into completely different categories: half-life of the precursor, activity of the precursor, trafficking machinery used by the precursor to leave the ER, activity of the precursor at the ER to name just a few speculative ideas. Therefore, the last sentences of the Conclusion section are strongly over-stated.

This work is solid and interesting. It presents a good set of figures. I don't perceive the manuscript as outstanding because the actual function of dPob is inferred from correlations with highly complex steady-state effects on selected individual proteins or on global ER homeostasis. From an outstanding paper I would expect either a systematic and unbiased approach to identifying the substrates of the putative chaperone complex or a delineation of the features that it recognizes in its substrates or some direct evidence for the actual chaperone activity proposed.

---

## [Author Response]

We made some changes regarding the reviewers’ comments.

Reviewer #1

*In*
Figure 5*, right panel it is not clear why the sec61::GFP input shows a band the size of the EMC1::GFP*; *what is the nature of this band?*

We think this band indicated a protein cross-reacting to anti-GFP antibody. We added the following sentence to the figure legend.

In both input extracts prepared from Rh1-Gal4/UAS-EMC1::GFP or sec61::GFP flies, there is the band with the same position to EMC1::GFP: this band will be the protein cross-reacting to anti GFP antibody

*Equally why is the +/- vitamin A experiment performed with different driver lines? wouldn't it be better to do both with the Rh1-Gal4 driver? Please give the rationale of using the two drivers*.

Since the overall expression level of EMC1::GFP was strong, hs-Gal4 driver was used to activate UAS:EMC1::GFP for the most of experiment. To analyze the interaction between EMC1 and Rh1-apoprotein, Rh1-Gal4 driver was also used because the expression of EMC1::GFP was stronger in the photoreceptors. We added this sentence in the result section.

Reviewer #3

*1) The authors show that EMC1*::*GFP does not co-immuno-precipitate with Rh1. Does calnexin co-IP with Rh1? I ask because the authors conclude (in the Discussion) that “ EMC is unlikely to be working in the calnexin cycle”, and it is difficult for me to understand why the authors have come to that conclusion*.

Indeed, it is reported that Cnx is co-immunoprecipitated with *Drosophila* Rh1 (Rosenbaum, 2006).

Our result indicated that EMC1-GFP co-IPs with Cnx. If EMC-Cnx complex work for Rh1 in calnexin-cycle, physical interaction is expected between ER-accumulated Rh1 apoprotein and EMC-Cnx complex. However, EMC1::GFP does not co-IP with Rh1, suggesting EMC-Cnx complex does not function in the calnexin-cycle but Cnx does.

In addition to the calnexin cycle, Cnx is also know to recognizes nascent polypeptides co-translationally in the ER lumen (Chen, 1995). We think it is more likely EMC-Cnx complex functions in this process.

*2) The authors report negative results with the ERAD component mutants, EDEM1, EDEM2 and VCP/+. I don't see the point of showing such negative data, as one cannot draw any conclusions. Perhaps the individual EDEM mutants do not show Rh1 phenotype as they are redundant (in mammals, EDEMs are known to be genetically redundant). Also, VCP's effect was assessed in a heterozygous condition, and a negative result here simply means that there is no dominant genetic effect. How about analyzing homozygous clones? Alternatively, since ERAD is not a major point of this work, the authors may want to take out this part, as it does not contribute to the overall story*.

Regarding the redundancy for EDEM, it is shown that RNAi knockdown of EDEM1 increases aggregation of P23H folding-mutant of human rhodopsin (Kosmaoglou, 2009). *Drosophila* has only two EDEM orthologs, EDEM1 and EDEM2 in the genome. We used mutant clones lacking dPob, EDEM1 and EDEM2, but did not see Rh1 apoprotein accumulation in the ER.

Regarding the heterozygous usage for VCP, It is shown that The degradation of Rh1 intermediate in Rh1[P37H] folding-mutant is restored by the heterozygous mutation of TER94^26-8^ (Griciuc, 2010).

However, we also agree with the reviewer’s opinion: these negative data does not draw clear conclusions. Therefore, we will take out this part.

[Editors’ note: the author responses to the previous round of peer review follow.]

*The reviewers noted the significance of your discovery of a role for a* Drosophila *member of the EMC complex in the maturation of rhodopsin. Your discovery that dPob/EMC3 acts upstream of ninaA in rhodopsin maturation was specifically credited with supporting a role for this member of the EMC complex in the early steps of the maturation of a multi-pass membrane protein. However, the expert reviewers were unanimous in their view that these finding do not add up to an advance of sufficient measure to merit publication in* eLife*.*

*It is possible that a different manuscript, for example one with further details on genetic lesions in other* Drosophila *EMC homologs or one that carried the biochemical analysis of the dPob/EMC3 lesion further would had received a warmer welcome at* eLife*. Unfortunately, the manuscript before us is too far off this mark to suggest specific experiments that would render it suitable for this journal.*

*Thus we are left with no choice but to return the manuscript to you in the hope that you will find the reviewers’ comments (appended below) of use as you prepare it for submission elsewhere*.

We are pleased to read reviewer’s constructive criticism and suggestion, and tried to answer them with sincere. Our revised manuscript includes three sets of new data, and three minor changes in figures, based on reviewer’s suggestion. Based on our new data, we slightly shifted our conclusion from the original manuscript, but mostly the same. We believe now our conclusion, “EMC is essential for biosynthesis of rhodopsin and other multi-pass membrane proteins in *Drosophila* photoreceptors”, becomes more solid and reliable.

Three sets of new data are following:

1) In our original manuscript, we analyzed only a subunit of EMS, dPob/EMC3. However, in this revised manuscript, in a large scale screening of EMS induced mutants deficient in Rh1 expression, we identified only two mutants with dPob-like phenotype. These two mutants were carrying loss of function mutations of EMC subunits: one is on EMC1, and the other is on EMC8/9, which yeast lacks. We analyzed the accumulation of immature Rh1 and substrate specificities in these mutants. In both loss of function mutants for EMC1 and EMC8/9, immature Rh1 fails to accumulate in ER (Figure 4), and the expressions of multi-pass transmembrane proteins, but neither a secreted nor type-I, II and IV single-pass transmembrane proteins are greatly reduced (Figure 7). The substrate specificities shown in the deficiencies of EMC1 and EMC8/9 are exactly same as that in dPob deficiency. This is the first functional study of the additional subunits of EMC, which yeast lacks.

2) The absence of Rh1 apoprotein can be explained by the degradation of misfolded Rh1 in the EMC null mutants through the accelerated ERAD pathway. Thus, we investigated if mutations on ERAD components restore immature Rh1 in ER. However, unlike in the two folding mutants of Rh1 (Kang and Ryoo, 2009; [19]; Griciuc et al., 2010), disturbance of ERAD activity did not restore expression of Rh1 in dPob/EMC3-deficient photoreceptors (Figure 9). Together with the epistasis over ninaA, the higher susceptibility of the Rh1 to the ERAD pathway than that of the two folding mutants of Rh1, imply that the EMC complex is more likely to be involved in the earlier processes such as membrane integration or co-translational folding than in the folding of fully translated, membrane integrated Rh1-apoprotein.

3) We performed co-IP experiment using EMC1::GFP as a bait, because dPob::GFP is not functional. We could confirm EMC1::GFP interacts with dPob, but we failed to show significant interaction between EMC1::GFP and immature Rh1 apoprotein in the ER (Figure 5). This result also supports the EMC complex is likely involved in the early biogenesis for multi-pass membrane proteins than in the folding of fully translated, membrane integrated Rh1-apoprotein.

By this co-IP experiment, we also found EMC1::GFP interacts with calnexin (Cnx). It has been reported that the photoreceptors of an amorphic mutant of Cnx show complete loss of Rh1 apoprotein (31) just as we showed in dPob, EMC1 or EMC8/9 mutant. Moreover, both Cnx and EMC are epistatic to the mutant of the rhodopsin-specific chaperon NinaA, which accumulates Rh1 apoprotein in the ER. These results indicate that EMC and Cnx can work together in Rh1 biosynthetic cascade prior to NinaA works.

Three minor changes in figures are following:

1) We add Rh1 staining of WT, ninaA^p263^ mutant, dPob^delta4^ mutant ommatidia (Figure 3).

2) This revised manuscript includes the analysis of three single-pass transmembrane proteins and one double-pass transmembrane protein, which we did not work for the original manuscript (Figure 6).

3) We removed Figure 9 in the original manuscript. We just described the results in the text.

Reviewer #1

*[…] Other findings, such as the altered ER morphology in mutant tissue and the activation of the UPR are less informative and more anticipated by the yeast work*.

*Thus the crucial issue for the reviewers is to establish the strength of the experimental data and, importantly the degree to which it advances our understanding of the EMC's biological role. If the application of this* Drosophila *genetic system has provided strong evidence favoring a selective role for this EMC component in early biogenesis of multi-pass transmembrane proteins, then the paper would be of interest to a broad readership regardless of the degree to which these findings were anticipated by prior speculation. If, however, the additional experimental data derived from the* Drosophila *system were deemed merely incremental, the paper would be more suited for a specialist journal.*

In this revised version of our paper, we investigated not only dPob/EMC3 deficiency, but also the deficiencies for other two subunits of EMC (see the answer for Reviewer#2 in detail). These two mutants were identified in a large scale screening of EMS-induced mutant deficient in Rh1 expression. Among 233 lines of Rh1-expression mutants, only two of them showed dPob-like phenotype (loss of Rh1 and NaK-ATPase but normal Eys expression), and these two were loss of function mutant of EMC subunits. Importantly, the deficiencies for three subunits of EMC gave the exactly same substrate specificity: they are essential for all of multi-pass transmembrane proteins, but not for a secreted protein or type-I, II and IV single-pass transmembrane proteins. Coincidence of the phenotypes in the deficiencies for three subunits of EMC provides the strong evidence favoring a selective role for this EMC component in early biogenesis of multi-pass transmembrane proteins.

Reviewer #2

*[…] Although the authors have done a solid job in their characterization of dPob function in* Drosophila*, it appears to be an extension of the* C. elegans *work (Richard, 2013), but with more analysis of potential substrates. Moreover, this study does not address many of the pressing questions regarding the EMC complex. Does dPob's role in* Drosophila *reflect the entire EMC complex function? What is the mechanistic basis of the specificity of dPob towards multispan membrane proteins? Based on these grounds, whether the overall novelty and scope of this study justifies the publication of* eLife *can be subject to debate. Below are a few specific comments along these lines*.

*1) The authors had identified dPob through a previously reported genetic screen (Satoh, 2013). Were there other EMC homologs identified in the screen? If it is the case, the authors should highlight those results, as it would indicate that the observed effect is not an isolated function of dPob*.

The previous screening among FRT-combined transposon insertion lines included only dPob/EMC3 among EMC subunits. The null mutant of dPob shows quite characteristic phenotype; no detectable protein expression of Rh1, and very weakened expression of other multiple-transmembrane domain proteins such as Na^+^K^+^-ATPase or TRP in the mosaic retina. We did not find any other mutant lines with such phenotype in the course of the mosaic screening among 546 insertional mutants described previously (38). To explorer other mutants showing phenotypes similar to dPob null mutant, we examined a collection of 233 mutant lines deficient of Rh1 accumulation in photoreceptor rhabdomeres, obtained in an ongoing ethyl methanesulfonate (EMS) mutagenesis screening. Among them, only two lines, 665G and 008J showed dPob-like phenotype in the mean of distribution of Rh1 and Na^+^K^+^-ATPase in the mosaic retina. 665G and 008J turned out to be frame-shift and nonsense mutations of EMC1 and EMC8/9, respectively. Thus, we included the results of analysis for these mutations to our revised manuscript.

*2) EMC genes are grouped because of their genetic interactions with each other in yeast. This study is potentially significant because the function of EMCs still remain poorly understood. Naturally, one must ask whether other EMC gene homologs show similar effects in rhodopsin maturation or not. The authors can examine this through in vivo RNAi, or better, by use of available mutant alleles*.

We found the null mutants of EMC1 and EMC7/8 show the phenotype which is characteristic to the dPob null mutant. Mutants of other subunits of EMC were not available. We did not perform RNAi experiment of other EMCs because the phenotype expected in RNAi experiment will be hypomorphic, and we knew the hypomorphic mutant of dPob shows just reduced Rh1 expression. Previously, we have tried RNAi screening of genes required for the Rh1 expression, and found out that too many genes show reduced Rh1 expression when derived by GMR-Gal4, probably because of the off-target effect, and none of them were phenocopied by null allele of the genes. Contrary, when RNAi was derived by Rh1-Gal4, the genes known to be required for Rh1 transport, showed no phenotype. Altogether, in *Drosophila* photoreceptor, we think RNAi experiments will not provide phenotype specific enough to conclude something.

*3) Does dPob physically interact with rhodopsin and other multipass membrane proteins? Immuno-precipitation experiments may be useful*.

We performed co-IP experiment using EMC1::GFP as a bait, because dPob::GFP did not localize to the ER. We could confirm EMC1::GFP interacts with dPob, but we failed to show the stable interaction between EMC1::GFP and Rh1 apoprotein accumulated in the ER in the VA-condition.

In addition, we found the mutations of ERAD components fail to restore Rh1 immature in dPob mutant cells. From these two results with epistasis over ninaA, we now think EMC complex is more likely to be involved in the earlier processes such as membrane integration or co-translational folding than in the folding of fully-translated, membrane-integrated Rh1-apoprotein.

Interestingly, we found Calnexin (Cnx) binds to EMC1::GFP in the co-IP experiment. The photoreceptors of an amorphic mutant of Cnx show complete loss of Rh1 apoprotein (Rosenbaum, 2006) just as shown in dPob, EMC1 or EMC8/9 mutant. Moreover, both Cnx and EMC are epistatic to the mutant of the rhodopsin-specific chaperon NinaA, which accumulates Rh1 apoprotein in the ER. These results indicate that EMC and Cnx can work together in Rh1 biosynthetic cascade prior to NinaA works.

*4) Is there any mechanistic insight as to how dPob would specifically recognize multipass membrane proteins? Although the authors show the dPob locus in*
Figure 1*, they do not describe any domain structures in the gene. The fact that dPob specifically affects transmembrane domain is an intriguing phenomenon, but a mechanistic insight is lacking*.

We now think EMC complex is more likely to be involved in the earlier processes such as membrane integration or co-translational folding, than in the folding of fully-translated, membrane-integrated Rh1-apoprotein (see above for the reason of this).

Most of the newly synthesized multi-pass membrane proteins are co-translationally integrated into the ER membrane through the translocon complex, and only one or two transmembrane helices can be stored in the translocon channel and the lateral gate. The helices associated with the translocon are displaced by the next set of newly synthesized transmembrane helices in the membrane-integration of multi-pass membrane proteins (Cymer, 2014; Rapoport, 2004). We assume that EMC may bind and stabilize the displaced helices, facilities the displacement, or involves to the integration, however, only sophisticated biochemistry with in vitro translation system can address the issue. We are applying for funding to start it, though we do not have enough resource to do it now.

*5) Does dPob physically interact with rhodopsins and other targets? A simple immune-precipitation experiment should provide answers*.

We answered to this question in point 3 above.

Reviewer #3

*[…] This work is solid and interesting. It presents a good set of figures. I don't perceive the manuscript as outstanding because the actual function of dPob is inferred from correlations with highly complex steady-state effects on selected individual proteins or on global ER homeostasis. From an outstanding paper I would expect either a systematic and unbiased approach to identifying the substrates of the putative chaperone complex or a delineation of the features that it recognizes in its substrates or some direct evidence for the actual chaperone activity proposed*.

Because of the limitation of our model system, we could not perform a systematic approach to identifying the substrates. Instead, we looked more substrates and we used more accurate topological classification for membrane proteins.

Now we do not think chaperon is the only possible function of EMC. We showed that weakening ERAD pathway does not suppress the loss of Rh1 by slowing down ERAD-dependent degradation. We also performed Co-IP with EMC1::GFP but did not detect stable binding with ER-accumulated Rh1 apoprotein. Our new results indicate it is more possible that EMC has some function on earlier steps than folding of fully translated protein, such as stabilizing translation, co-translational folding or membrane integration.